# Single cell transcriptomics shows that malaria promotes unique regulatory responses across multiple immune cell subsets

Nicholas L. Dooley [1,2,9], Tinashe G. Chabikwa[1,9,10], Zuleima Pava[1], Jessica R. Loughland[1], Julianne Hamelink [1,3], Kiana Berry[1,4], Dean Andrew[1], Megan S. F. Soon[1], Arya SheelaNair[1], Kim A. Piera [5], Timothy William[6,7], Bridget E. Barber [1,5,6], Matthew J. Grigg [5,6], Christian R. Engwerda[1], J. Alejandro Lopez [1,2], Nicholas M. Anstey[5,6] & Michelle J. Boyle [1,2,3,4,8] ✉

*Plasmodium falciparum* malaria drives immunoregulatory responses across multiple cell subsets, which protects from immunopathogenesis, but also hampers the development of effective anti-parasitic immunity. Understanding malaria induced tolerogenic responses in specific cell subsets may inform development of strategies to boost protective immunity during drug treatment and vaccination. Here, we analyse the immune landscape with single cell RNA sequencing during *P. falciparum* malaria. We identify cell type specific responses in sub-clustered major immune cell types. Malaria is associated with an increase in immunosuppressive monocytes, alongside NK and γδ T cells which up-regulate tolerogenic markers. IL-10-producing Tr1 CD4 T cells and IL-10-producing regulatory B cells are also induced. Type I interferon responses are identified across all cell types, suggesting Type I interferon signalling may be linked to induction of immunoregulatory networks during malaria. These findings provide insights into cell-specific and shared immunoregulatory changes during malaria and provide a data resource for further analysis.

*Plasmodium falciparum* causes significant disease burden globally, with >240 million cases of malaria and >600,000 deaths reported in 2021[1]. In areas of high malaria transmission children rarely experiencing recurrence of severe malaria due to the rapid acquisition of anti-disease or tolerogenic immunity[2], and/or antibody mediate protection from severe disease-causing parasite strains[3]. However, anti-parasitic immunity and protection from mild disease develops more slowly, with children experiencing multiple patent, symptomatic infections throughout childhood before developing levels of protection that control parasite growth to sub-patent levels[4]. These phenomena are thought to be linked, with the slow acquisition of anti-parasitic immunity attributed to tolerogenic responses across multiple cell subsets required for robust adaptive immune development. These tolerogenic mechanisms may also contribute to reduced malaria vaccine efficacy in exposed populations[5]. As such, a better understanding of tolerogenic immune responses during infection may inform the

[1]QIMR Berghofer Medical Research Institute, Brisbane, QLD, Australia. [2]School of Environment and Sciences, Griffith University, Brisbane, QLD, Australia. [3]University of Queensland, Brisbane, QLD, Australia. [4]Queensland University of Technology, Brisbane, QLD, Australia. [5]Menzies School of Health Research, Charles Darwin University, Darwin, NT, Australia. [6]Infectious Diseases Society Kota Kinabalu Sabah-Menzies School of Health Research Program, Kota Kinabalu, Sabah, Malaysia. [7]Subang Jaya Medical Centre, Selangor, Malaysia. [8]Burnet Institute, Melbourne, VIC, Australia. [9]These authors contributed equally: Nicholas L. Dooley, Tinashe G. Chabikwa. [10]Deceased: Tinashe G. Chabikwa. ✉e-mail: michelle.boyle@burnet.edu.au

development of more effective vaccine strategies for areas of high malaria endemicity.

Malaria driven tolerogenic and immunoregulatory responses allow parasite persistence by evading and disrupting anti-parasitic mechanisms employed by both innate and adaptive immune cells. Multiple studies have shown that monocytes and dendritic cells (DCs), that are initiators of the immune responses, are tolerized during malaria. During experimental human malaria and natural infection, monocytes and DCs have reduced responsiveness to toll-like receptor (TLR) stimulation and antigen presentation is suppressed[6-9]. Increased IL-10 production[10] and higher frequencies of monocytes with a regulatory phenotype are also detected in children and adults from malaria endemic areas[11,12]. Tolerogenic phenotypes have also been reported in other innate cells. For example, natural killer (NK) cells expressing the regulatory marker PD1, are expanded in populations in malaria endemic areas[13]. Additionally, gamma-delta (γδ) T cells, particularly Vδ2+ subsets which are important innate inflammatory responders to malaria parasites, become tolerized in children in high transmission settings[14,15]. Immunoregulation also exists in adaptive cell responses to malaria. Within the CD4 T cell compartment, type 1 regulatory (Tr1) cells that co-produce IFNγ and IL-10 during malaria, dominate antigen-specific CD4 T cell responses in children in high endemic areas[16-18]. These Tr1 cells develop rapidly during a primary malaria infection in previously naive adults[19]. CD4 T cells also upregulate additional inhibitory pathways during malaria infection, including expression of multiple co-inhibitory receptors and production of TGFβ[20-22]. Within the B cell compartment, multiple studies have shown that atypical memory B cells expand in response to malaria[23-25]. These cells have reduced functional capacity compared to 'typical' memory B cells[26,27], and an immunoregulatory role for these cells in malaria is also possible. Malaria responsive regulatory B cells (Bregs), which produce IL-10, have been reported in mouse models[28], but have not been identified in human malaria. While the drivers of tolerogenic cell responses are incompletely understood, Type I IFN signalling is key to the emergence of Tr1 CD4 T cells during malaria[19], and is recognised as both an activating and regulatory driver of the malaria immune response[29].

Transcriptional changes associated with immune cell tolerance have been reported in limited studies. For example, monocytes from Malian children and adults following parasite stimulation in vitro had reduced induction of NFKB1 (positive regulator of inflammation) in tolerized adult cells[11], and transcriptional analysis of Vδ2+ γδ T cells in Ugandan children identified upregulation of multiple immunoregulatory pathways in highly exposed children[14]. Additionally, a large whole-blood transcriptomic study revealed the upregulation of interferon responses, and that p53 activation in monocytes attenuated Plasmodium-induced inflammation and predicted protection from fever[30]. However, to date, no studies have comprehensively investigated transcriptional signatures of malaria-driven tolerance across all cell subsets in the same individuals during infection.

The advent of single-cell RNA sequencing (scRNAseq) has allowed comprehensive analysis of distinct immune cell subsets during human infection, and identification of key changes driven by infection. To date, while scRNAseq has been applied to the malaria parasites[31-33], and to immune cells in a single experimental infection study[34], no comprehensive scRNAseq mapping of the immune landscape during malaria infection has been undertaken. In the present study, we applied scRNAseq to peripheral blood mononuclear cells (PBMCs) from patients during acute *P. falciparum* malaria and post treatment. Cell linages and subsets were identified, differential expressed genes (DEGs) associated with acute malaria infection compared to convalescence analysed and key transcriptional changes confirmed at the protein level in additional patients. Together this study advances our understanding of the regulatory immune landscape during malaria

and provides opportunities to manipulate these pathways for clinical advantage.

## Results

### Altered immune cell profiles during acute malaria infection

To undertake a global analysis of the immune response during malaria infection, we performed droplet-based scRNAseq on peripheral blood mononuclear cells (PBMCs) from 6 individuals (age 6–24 years) with uncomplicated *P. falciparum* malaria at enrolment (day 0, acute malaria time point) and at 7 and 28 days after drug treatment, along with 2 healthy adult endemic controls (*EC*, ages 20 and 27 years) (Fig. 1a, Supplementary Table 1). We sequenced a total of 115,526 cells, with 106,076 cells passing quality control (QC; minimum of 220 genes expressed and <20% mitochondrial reads per cell) (Supplementary Fig. 1A). Due to a 10X Chromium wetting error, no quality cells were retained from one individual at the acute infection time point (child1day0, Supplementary Fig. 1B). Data were integrated to harmonize data sets across batch, donor and infection timepoints, and cell clusters visualized with uniform manifold approximation and projection (UMAP) (Supplementary Fig. 1C). Expression of canonical and lineage marker genes were used to annotate cell clusters into 15 high level cell states: CD14+ classical monocytes, CD16+ non-classical monocytes, classical dendritic cells (cDCs), plasmacytoid dendritic cells (pDCs), CD4 T cells, CD8 T cells, γδ T cells, NKT cells, B cells, plasma cells, proliferating [which appeared to be of mixed cell types], hematopoietic stem and progenitor cells (HSPCs), platelets, and one uncharacterized cluster (Fig. 1b/c, Supplementary Data 1, Supplementary Fig. 1D). The uncharacterized cluster did not express any known linage marker genes, and had relatively increased expression of mitochondrial genes, possibly indicating low quality cells. The relative proportions of identified cell clusters correlated strongly with the proportions of cells identified by flow cytometry analysis of the same cell samples (R = 0.95, p < 0.001, Fig. 1d, Supplementary Fig. 2). During malaria infection, there were some changes to the distribution of cell types, with trends of increases in CD4 T cells (p = 0.059 day 0 v day 28) and proliferating cells (p = 0.03 day 0 v day 28, and marked decreases in NK cells (p = 0.024 day 0 v day 7), γδ T cells (p = 0.035 day 0 v day 28) and B cells (p = 0.026 day 0 v day 28), as a proportion of total analyzed cells within each individual (Fig. 1e). To characterize gene expression profiles during malaria infection, we performed differential gene expression analysis within each cell subset between acute infection (day 0), and 7 and 28 days after treatment (Fig. 1f). We observed the largest number of DEGs when comparing between day 0 and day 28, with monocytes and cDCs exhibiting the highest transcriptional changes relative to other cell types (Fig. 1f, Tables S3A–C). Large numbers of DEGs were also detected between day 0 and day 7, with a large proportion of these also detected 28-days post-treatment for each subset (for example, majority of DEGs for day 0 compared to day 7 and day 0 compared to day 28, were shared for CD14+ classical monocytes [64%], CD16+ non-classical monocytes [60%], cDCs [53%] and pDCs [47%]). DEGs were also detected between day 28 (convalescence) and endemic controls, possibly indicated sustained changes to cell transcriptional profiles up to a month after infection, or alternatively, due to the different baseline transcriptional profiles between individuals (Supplementary Fig. 1E). DEGs were not calculated for each individual for each cell cluster due to low number of contributing cells for some individuals once subsets were identified (Supplementary Table 2). Subsequently, to identify DEGs caused by malaria infection compared to convalescence, we focused analysis on DEGs identified between day 0 (16,192 cells) and day 28 (34,286 cells).

### Shared and subset-specific immunosuppressive signatures in monocytes and cDCs during malaria

We first analysed transcriptional changes to innate myeloid cells comparing day 0 to day 28 and identified 1674, 1182, 521 DEGs in CD14+

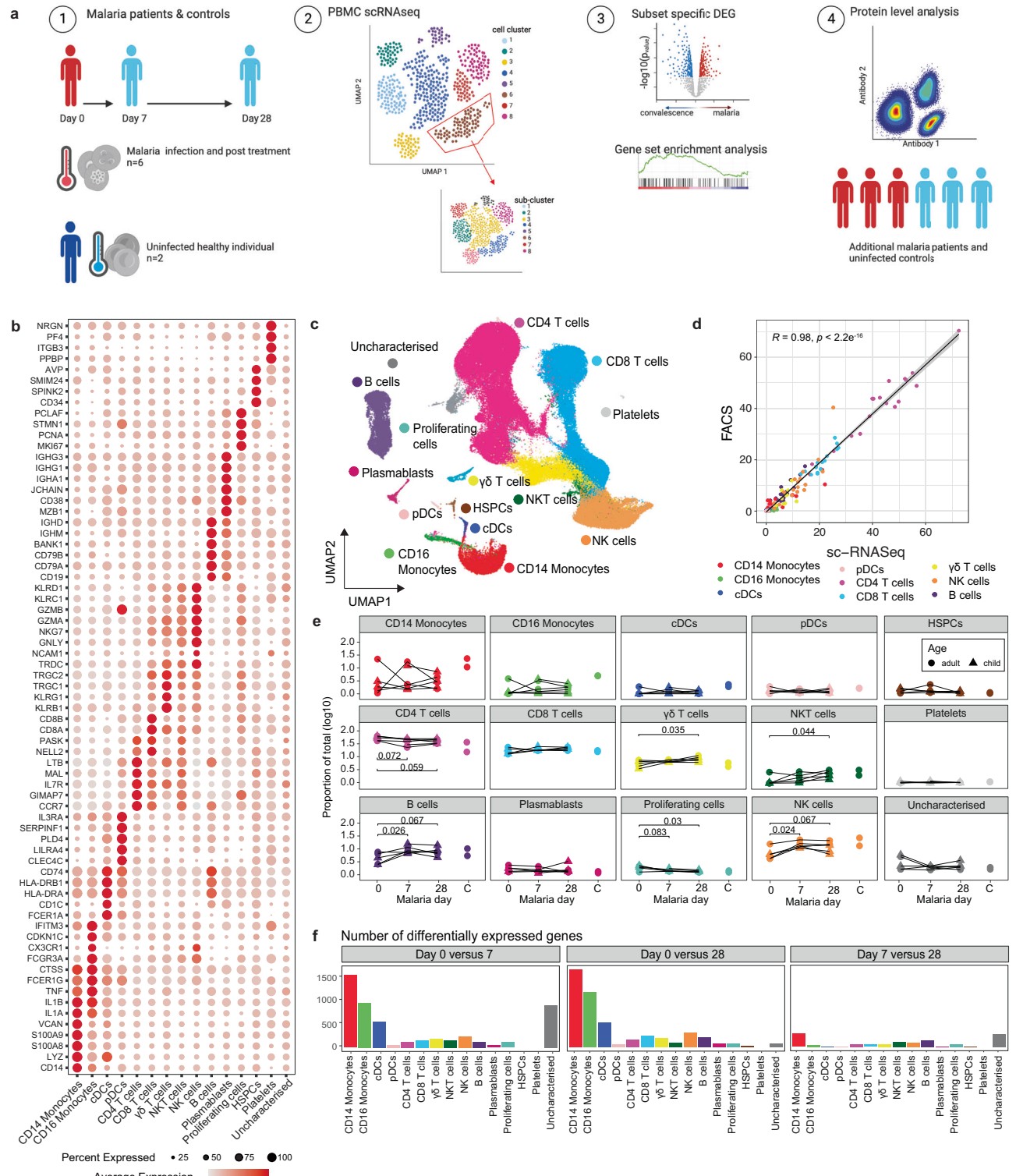

**Fig. 1 | Single cell transcriptional landscape of malaria infection. a** A schematic outline depicting workflow for sample collection and scRNAseq analysis. PBMCs were collected from *P. falciparum* malaria patients at day 0 ($n = 5$), and at day 7 ($n = 6$) and 28 ($n = 6$) post treatment and from healthy endemic controls without malaria (EC, $n = 2$). Live PBMCs were analysed by 3' 10X Chromium single cell sequencing, and cell types identified. For each cell type and sub-cluster, genes with differential expression between days were identified and analysed. Key findings were confirmed at the protein level in additional patients. Created with BioRender.com. **b** Dot plot of the mean expression of marker genes used to annotate cell types. **c** UMAP of all cells in integrated analysis. Cells are coloured by cell subtypes. **d** Correlation between relative proportion of cells identified by scRNAseq

and flow cytometry analysis (data from $n = 19$, from day 0 $n = 5$, day 7 $n = 6$, day 28 $n = 6$, endemic controls $n = 2$). Pearson's R and $p$ (two-sided) is indicated. Error is 95% confidence interval. **e** Relative proportion of identified subsets from scRNAseq analysis as a proportion of total cells analysed per timepoint/individual at day 0 during malaria, and day 7-, and 28-days post-treatment, and in healthy endemic control individuals. *P*-value (two-sided) is calculated by Mann Whitney *U* test between day 0 and subsequent time points. **f** Number of DEGs for each cell type between day 0/7, day 0/28 and day 7/28. See also Supplementary Fig. 1-2 and Supplementary Tables 1-2 and Supplementary Data 1-2. Source data are provided as a Source Data file.

classical monocytes, CD16+ non-classical monocytes and cDCs, respectively (Supplementary Data 2). DEGs were identified by comparing all cells at day 0 to day 28 in each cell cluster, regardless of individual donor. However, for all DEGs, the changes to expression between day 0 and day 28 were largely consistent across individuals (Supplementary Fig. 3). The high transcriptional activity of monocytes and cDCs during malaria contrasted with the low number of DEGs detected in pDCs ($n = 60$). Of pDC DEGs, 23% were associated with the long non-coding RNA family and 15% were histone genes (Supplementary Data 2). Low transcriptional activation of pDCs is consistent with our previous bulk-RNA sequencing analysis of isolated pDCs during experimental malaria infection, which showed that pDCs were transcriptionally stable during infection[35]. Transcriptional changes during malaria in both monocyte subsets and cDCs were both shared and cell type specific (Fig. 2a). Both CD14+ and CD16+ monocytes were activated, with marked upregulation of innate cell activation genes such as Toll-like Receptors (TLRs), disintegrin and metallopeptidase domain (*ADAM9* and *ADAM10*) proteins[36], as well as alarmins, *S100A8* and *S100A9*, which are typically upregulated in monocytes under inflammatory conditions[37]. In contrast, MHC class II HLA-DR genes were down-regulated in both CD14+ and CD16+ monocytes (Fig. 2b, Supplementary Data 2). High expression of *S100A8/A9*, along with *RETN*, *ALOX5AP* and reduced expression of MHC class II HLA-DR genes has recently been used to define immunosuppressive MS1 monocytes in sepsis[38] and proportional increases in these immune-suppressive monocytes has been detected in both sepsis and COVID-19[39], however these immunosuppressive monocytes have not been identified in other acute infections such as dengue[40]. Consistent with an enrichment of immunosuppressive monocytes during malaria infection, inflammatory cytokine genes including *TNF, IL1α, IL1β, IL6, IL18* were markedly reduced at day 0, compared with 28 days post treatment (Fig. 2b, Supplementary Data 2). There was also reduced expression of multiple chemokine genes including *CCL2* (encoding MCP-1), *CCL3, CCL4, CCL5, CCL7, CXCL8* (encoding IL8), *CXCL2, CXCL3, CCL20, CXCL1* and *CXCL16* in both monocyte subsets, with a greater magnitude of reduction in CD14+ monocytes. Consistent with reduced expression of these cytokine and chemokine genes, we also observed reduced expression of NK-κB family members, *NF-κB1, NF-κB2,* and *REL*, central transcriptional factors for pro-inflammatory gene induction[41]. Immune-suppressive phenotypes were also detected in cDCs during infection, with notable downregulation of HLA-DR genes *HLA-DRA and HLA-DRB1* along with multiple paralogues (*HLA-DPA1/B1* and *HLA-DQA1/B1*) in cDCs but not cytokine/chemokine genes. The reduction of HLA-DR genes in cDCs is consistent with previous reports showing reduced HLA-DR expression at the protein level on DCs during experimental[6], and naturally acquired malaria[8,42,43].

In contrast to down-regulation of inflammatory gene signatures, multiple genes involved in pathogen recognition, scavenging and phagocytosis were upregulated in both monocyte subsets and cDCs during acute infection, including *CD163* and *FCGR1A* (encoding CD64/FcγRI) which we have previously shown to be upregulated at the protein level on both classical and non-classical monocytes during malaria[7] (Fig. 2b). Similarly, *FCAR* (encoding the Fc fragment of IgA receptor) was also upregulated on both monocyte subsets and cDCs, consistent with a possible role of IgA antibodies in immunity against malaria[44–46]. *CD93* and *FCGR2A* (encoding CD32/FcγRIIa) which mediate the enhancement of phagocytosis in monocytes and macrophages were upregulated during infection on CD16+ monocytes and cDCs, but not CD14+ monocytes. *COMPLEMENT RECEPTOR 1* (*CR1*), a membrane immune adherence receptor that plays a critical role in the capture and clearance of complement-opsonized pathogens, was upregulated on both CD14+ and CD16+ monocytes and cDCs during malaria. This transcriptional upregulation of *CR1* is in contrast to previous reports of down regulated CR1 on splenic monocytes/macrophages in murine models of malaria, and on CD16+ monocytes/macrophages in *P.*

*falciparum* and *P. vivax* malaria patients from Peru[47]. *CD36*, involved in antibody independent phagocytosis was also upregulated on both subsets and cDCs. The downregulation of cytokine and chemokine responses in the myeloid compartment, but upregulation of receptors involved in antibody mediated parasite clearance during malaria is supported by recent work by others[48].

To further investigate shared and cell type specific transcriptional changes, DEGs were analysed using the Gene Set Enrichment Analysis (GSEA) and overrepresented upstream regulators identified. These analyses revealed both shared and cell specific gene signature enrichment and regulators (Fig. 2c, d). Consistent with DEGs for each subset, both monocyte subsets and cDCs were enriched for pathways such as secretion, phagocytosis, myeloid leukocyte mediated immunity and immune effector processes during malaria. In contrast, multiple subset specific pathways were also identified. For example, leukocyte migration and chemotaxis, inflammatory and defence response, and cytokine production were enriched at day 0 during malaria in CD16+ monocytes and cDCs but enriched at day 28 in CD14+ monocytes. In contrast, pathways associated with antigen presentation were consistently enriched only in CD16+ monocytes during malaria. In agreement with cell type specific pathway enrichment, analysis of upstream regulators identified motifs that were shared between CD14+ and CD16+ monocytes including *NFκB-p5, IRF2, STAT5* and *JUN-AP1*-binding *cis*-elements, but also regulators that were specific for each subset. For CD16+ monocytes, this included *p53, STAT1, IRF1,* and *IRF8*-binding *cis*-elements. p53 was previously identified in bulk transcriptional analysis as an important contributor of innate cell responses and immunity to malaria[30]. The enrichment of *IFR1/8/2* and *STAT1/5* are consistent with a key role of Type I IFN pathways in both activation and regulation of innate immune cell responses in malaria[29].

To assess some of these key findings at the protein level, ex vivo secretion of cytokine/chemokines TNF, IL-1β, IL-6 and MCP-1 (*CCL2*) from CD14+ monocytes, CD16+ monocytes and cDCs were measured in additional *P. falciparum* malaria patients from the same study site at day 0 and day 28 ($n = 15$) (Fig. 2e–g, Supplementary Fig. 4A, B, Supplementary Table 1). Consistent with transcriptional findings, the majority of individuals had reduced inflammatory cytokine/chemokine secretion in CD14+ monocytes at day 0 compared to day 28 (Fig. 2e, f). This reduction was significant at the population level for TNF and MCP1. MCP1 expression was also reduced at day 0 compared to day 28 in cDCs. Cytokine production was not associated with age/parasitemia or sex (Supplementary Fig. 4C–E). To measure the co-expression of cytokines, responses were analysed by SPICE (Simplified presentation of incredibly complex evaluations[49]), and cells expressing 0, 1, 2, 3, or 4 cytokines in any combination calculated. CD14+ monocytes were significantly less polyfunctional at day 0 compared to day 28 (Fig. 2g). Collectively, data reveal a significant enrichment of regulatory innate cells during malaria infection which down-regulate multiple cytokine and chemokine responses along with HLA-DR associated genes and pathways required for robust inflammatory control and antigen presentation in a cell subset specific manner, regulated by cell subset specific pathways. In contrast, multiple receptors involved in antibody mediated functions are upregulated, consistent with a potential role of innate cells in antibody mediated parasite clearance during infection[48].

## Subset-specific activation and regulation of NK cells during malaria

Along with innate myeloid cells, NK cells are important early responders to *Plasmodium* infection, but also have roles in adaptive immunity as effector cells. Broadly, NK cells exist in multiple distinct functional subsets along a spectrum of least differentiated CD56bright cells, towards highly differentiated CD56dimCD16+CD57+ senescent cells. CD56bright NK cells and other more differentiated subsets produce IFNγ following parasite stimulation in vitro[50,51] and during controlled human

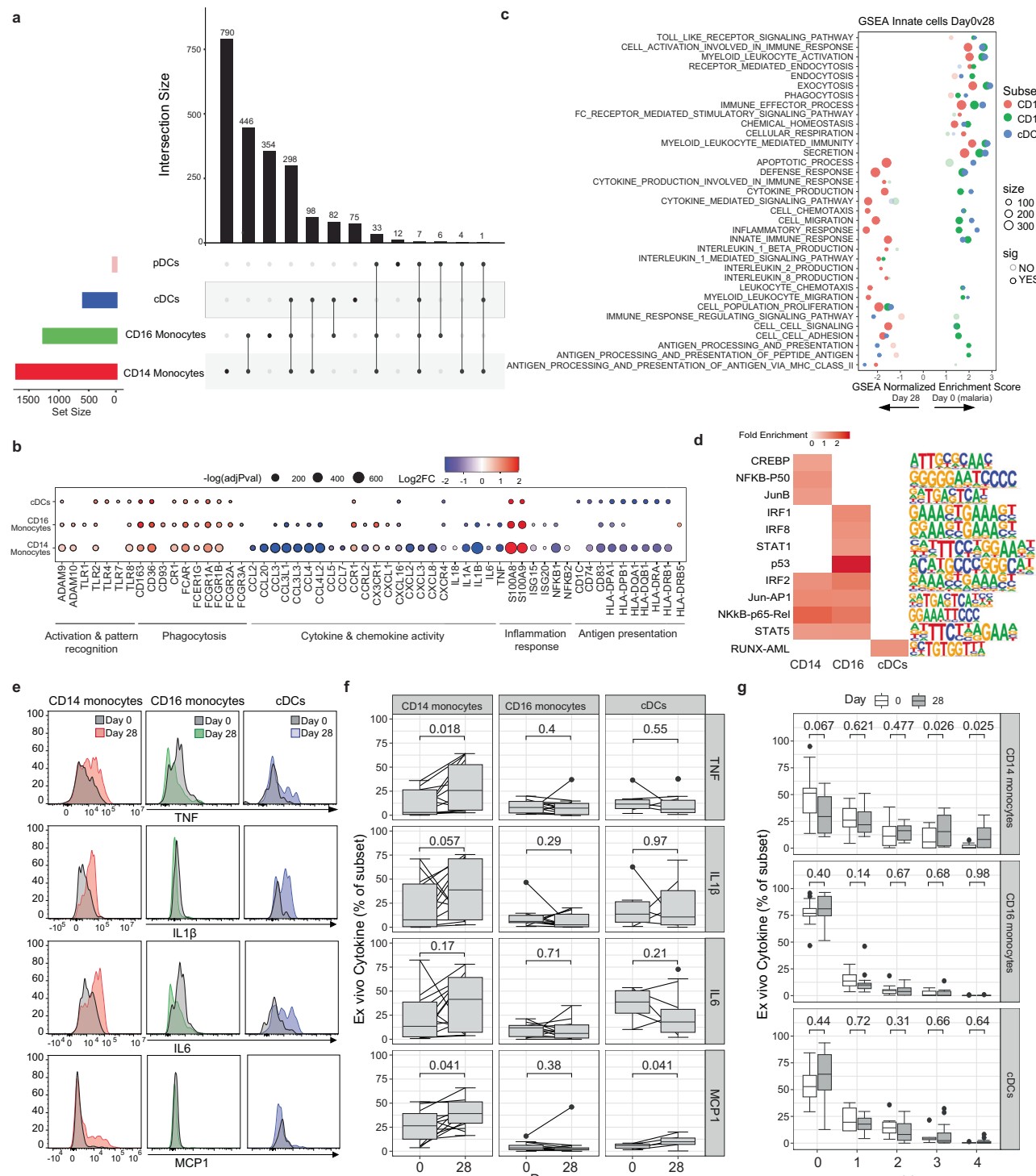

**Fig. 2 | Immuno-suppressive signatures of innate cells during malaria.** DEGs in CD14⁺ and CD16⁺ monocytes, and cDCs and pDCs of day 0 compared to day 28 were identified (*n* = 5 day 0, *n* = 6 day 28). **a** Upset plot of shared and subset specific DEGs of each subset identified between day 0 and day 28. **b** DEGs of interest in monocytes and cDCs. Genes with known monocyte function are indicated. Log transformed *p* values (Wilcox test, two-sided, Bonferroni correction). **c** GSEA of DEGs in CD14⁺ and CD16⁺ monocytes and cDCs. **d** Common and unique upstream regulators of DEGs in CD14⁺ and CD16⁺ monocytes and cDCs. **e** Ex vivo secretion of TNF, IL1β, IL6 and MCP1 was measured in CD14⁺ and CD16⁺ monocytes, and cDCs. Representative cytokine expression of a single individual at day 0 compared to day 28 in

each subset. Subset gates from each sample were excluded if <50 events (CD14⁺ monocytes – Day 0 *n* = 15, Day 28 *n* = 14; CD16⁺ monocytes – Day 0 *n* = 14, Day 28 *n* = 14; cDCs – Day 0 *n* = 6, Day 28 *n* = 14). **f** Cell subset level expression of cytokines. Box and whisker plots indicate first and third quartiles for the hinges, median line, and lowest and highest values no further that 1.5 interquartile range from the hinges for whisker lines. **g** Frequency of cell subset expressing none or a combination of the 4 cytokines: TNF, IL1β, IL6 and MCP1. *P*-value (two-sided) indicated is calculated by Mann-Whitney *U* test. See also Supplementary Fig. 3-4 and Supplementary Tables 1-2, Supplementary Data 1-2. Source data are provided as a Source Data file.

malaria infection[52]. In contrast, adaptive and highly differentiated NK cells expand in malaria exposed individuals, and function via antibody dependent cellular cytotoxicity (ADCC) to protect from malaria[53,54].

To investigate the transcriptional activation of these phenotypically distinct NK cell populations during malaria, we performed sub-clustering of NK cells identified in PBMCs (Fig. 1). Sub-clustering identified five subsets which were annotated based on cluster markers as CD56^bright, Transitional, IFNγ^+ Adaptive, IFNγ^- Adaptive and PD1^+ NK cell subsets (Fig. 3a, Supplementary Data 3). CD56^bright cells expressed the highest levels of *NCAM1* (encoding CD56), *SELL, KLRF1, GZMK* and *IL7R*. The Adaptive cell subsets had increased expression of *NKG7, GZMB* and *FCGR3A* (encoding CD16). Transitional NK cells expressed markers from both CD56^bright and Adaptive subsets, consistent with previous scRNAseq analysis[55,56](Fig. 3b). Within the Adaptive cell subsets, clusters were differentiated as IFNγ^+ and IFNγ^- Adaptive subsets based on *IFNG* expression, along with *TNF, CCL3* and *CCL4*. Additionally, our unsupervised clustering isolated an NK cell cluster expressing high levels of *PDCD1* (encoding PD1). PD1^+ NK cells have previously been shown to expand with age in malaria endemic populations and have increased ADCC function[13]. PD1^+ NK cells also expressed genes related to NK recruitment and activation, *VCAM1, ITGAD* (encoding CD11d), *TOX, TNFRSF1B* (encoding TNFRII) and *CD160* (Fig. 3b). During malaria, there was a significant increase in the proportion of IFNγ^+ Adaptive cells, consistent with an increased inflammatory and cytokine responsiveness of Adaptive NK cells during infection. Additionally, there was a proportional decrease in PD1^+ NK cells during acute infection compared to 7 days following treatment (Fig. 3c). This decreased proportion of PD1^+ NK cells during acute malaria is in contrast to previous reports of the expansion of this subset identified by flow-cytometry in Malians with *P. falciparum* malaria[13].

To investigate malaria-driven transcriptional changes to NK cells, we identified DEGs from day 0 to day 28 within each subset (Supplementary Data 4). While individuals contributed different cell numbers to each cluster, DEG changes were consistently expressed across individuals (Supplementary Table 3, Supplementary Fig. 5). Transitional, IFNγ^- Adaptive and PD1^+ subsets were more transcriptionally active during malaria compared to CD56^bright and IFNγ^+ Adaptive subsets (Fig. 3d). PD1^+, Transitional and IFNγ^- Adaptive NK cells, had a large proportion of subset specific DEGs, suggesting unique NK cell subset specific activation pathways during malaria (subset specific DEGs 76%, 42% and 47% respectively, Fig. 3d). During acute malaria, there was evidence for increased cytotoxic potential across multiple NK cell subsets, with upregulation of genes with known cytotoxic functions (Granzyme members *GZMB, GZMA, GZMK*[57], Granulysin *[GNLY]*, Perforin *[PRF1]*) and NK cell activation/degranulation (*CD44*[58], *CCL4L2*[59], *STX11* [encoding Syntaxin 11[60]], *CD8A*[61], *LGALS1* [encoding Galectin 1[62]], *XCL1, SLC7A5* [encoding CD98/LAT1[63]] and *SELL* [encoding CD62L[64]]) (Fig. 3e). GSEA confirmed upregulation of multiple pathways associated with cell function, particularly within Transitional, IFNγ^- Adaptive and IFNγ^+ Adaptive subsets (Supplementary Fig. 6). In PD1^+ NK cells, there was also evidence for subset specific upregulation of the MAPK pathway (increased *MAPKAPK2* and *MAP2K2*). Additionally, PD1^+ NK cells had a large increase in *ITGAX* (encoding CD11c) expression during acute infection, with smaller increases in *ITGAX* in CD56^bright and Transitional NK cells. CD11c expression in NK cells is upregulated in response to inflammatory cytokines[65], and is a marker of bitypic NK cells which can produce inflammatory cytokines and also drive the proliferation of γδ T cells[66]. Further, upregulation of HLA-DR genes *HLA-DQB1* and *HLA-DRB5* were also detected, with HLA-DR previously associated with NK cell activation and antigen-presentation in some settings[67].

Upregulation of genes mediating increased cytotoxicity and function during malaria was balanced by increases in multiple negative regulators of NK cells, including *TNFRSF4* (encoding CD134/OX40[68]), *TNFRSF9* (encoding CD137/4-1BB[69]), *TNFRSF18* (encoding CD357/

GITR[70]), *LAG3*[71], *HAVCR2* (encoding TIM-3[72]) and *CBLB*[73]. GSEA showed significant enrichment of negative regulatory pathways in PD1^+ and IFNγ^- Adaptive subsets ('negative regulation of immune system process' and 'negative regulation of lymphocyte activation') (Fig. 3e and Supplementary Fig. 7). Across all subsets, multiple Type I IFN signalling genes were upregulated including *ISG20, IFI16, IFI6, IFITM3, IRF4, IRF8*, and *IRF7* and GSEA confirmed upregulation of 'response to Type I IFN' in Transitional and IFNγ^- Adaptive cells (Supplementary Fig. 6). Type I IFN signalling in NK cells has been shown to suppress IFNγ production during viral infection[74]. Taken together data suggest that Type I IFN signalling is activated in NK cells in response to malaria, which may be linked to the induction of regulatory NK cells during infection.

To investigate these key transcriptional changes at the protein level, we analysed NK cells by flow cytometry in additional *P. falciparum* malaria patients (day 0) (*n* = 11) and patients 28 days after infection (*n* = 17). NK cells were identified as CD56^bright, Adaptive and PD1^+ subsets[13](Supplementary Fig. 7A-B). Within these patients, there was a slight increase in CD56^bright cells within the NK compartment during infection (Supplementary Fig. 7C). Across all three NK subsets, there was a significant increase in Granzyme-B and Perforin expression at day 0 compared to day 28 post treatment (Fig. 3f). In contrast, CD98 and HLA-DR expression were increased on Adaptive and PD1^+ NK cells, but not CD56^bright NK cells, and expression levels were much higher on PD1^+ cell subset. Similarly, the regulatory marker LAG-3 was increased in expression on all NK cell subsets at day 0 (at very low levels on CD56^bright cells), but TIM-3 was only increased on Adaptive and PD1^+ cells (Fig. 3f). Expression of activation and regulatory markers on NK cell subsets was not associated with parasitemia, age or sex, aside from a relatively weak association between CD98 expression on PD1^+ NK cells with parasitemia (rho = 0.4, *p* = 0.037) (Supplementary Fig. 8A, 9A, 10A). IFNγ and TNF expression were also assessed. While ex vivo IFNγ expression was below the limit of detection for all NK cell subsets, there was a significant increase in TNF production in Adaptive, but not CD56^bright NK cells during malaria (Supplementary Fig. 7D). Additionally, there was a significant increase in CD11c expression on Adaptive NK, but not CD56^bright cells (Supplementary Fig. 7E). When considering the total profile of Granzyme-B, Perforin, CD98, HLA-DR, LAG-3 and TIM-3 on the three distinct NK cell subsets, there was a significant difference in the overall composition of marker expression in each subset between day 0 and day 28, indicating significant upregulation of both cytotoxic and regulatory markers during infection (Fig. 3g). However, PD1^+ NK cells are the most highly activated and regulatory during malaria due to the increased overall level of expression of these markers during acute infection, particularly CD98, HLA-DR, LAG-3 and TIM-3, compared to adaptive and CD56^bright NK cells (Fig. 3h, Supplementary Fig. 7F).

## Activation and regulation of γδ T cells with diverse functions during malaria

γδ T cells are key innate cell responders during malaria which proliferate in response to malaria parasites and produce inflammatory cytokines with important roles in protection[75]. γδ T cells can also recognize and kill parasites via lysis, and opsonic phagocytosis[76], and present antigen to activate T cells[77]. However, in individuals who have had repeated malaria infections, γδ T cell activation is highly regulated, with reduced cell frequency, reduced inflammatory responsiveness and increased expression of proteins related to γδ T cell immunoregulation and exhaustion[14,78]. γδ T cells with increased expression of regulatory markers are associated with protection from symptomatic disease and, also have increased CD16 expression and CD16-dependent cytolytic responsiveness to opsonized parasites[14,79]. To explore these multiple roles and tolerization mechanisms of γδ T cells in the patients with acute malaria in a low endemic setting in this study, we sub-clustered γδ T cells identified in PBMCs (Fig. 1) and categorized 6 clusters as Cytotoxic, Inflammatory, Antigen-presenting,

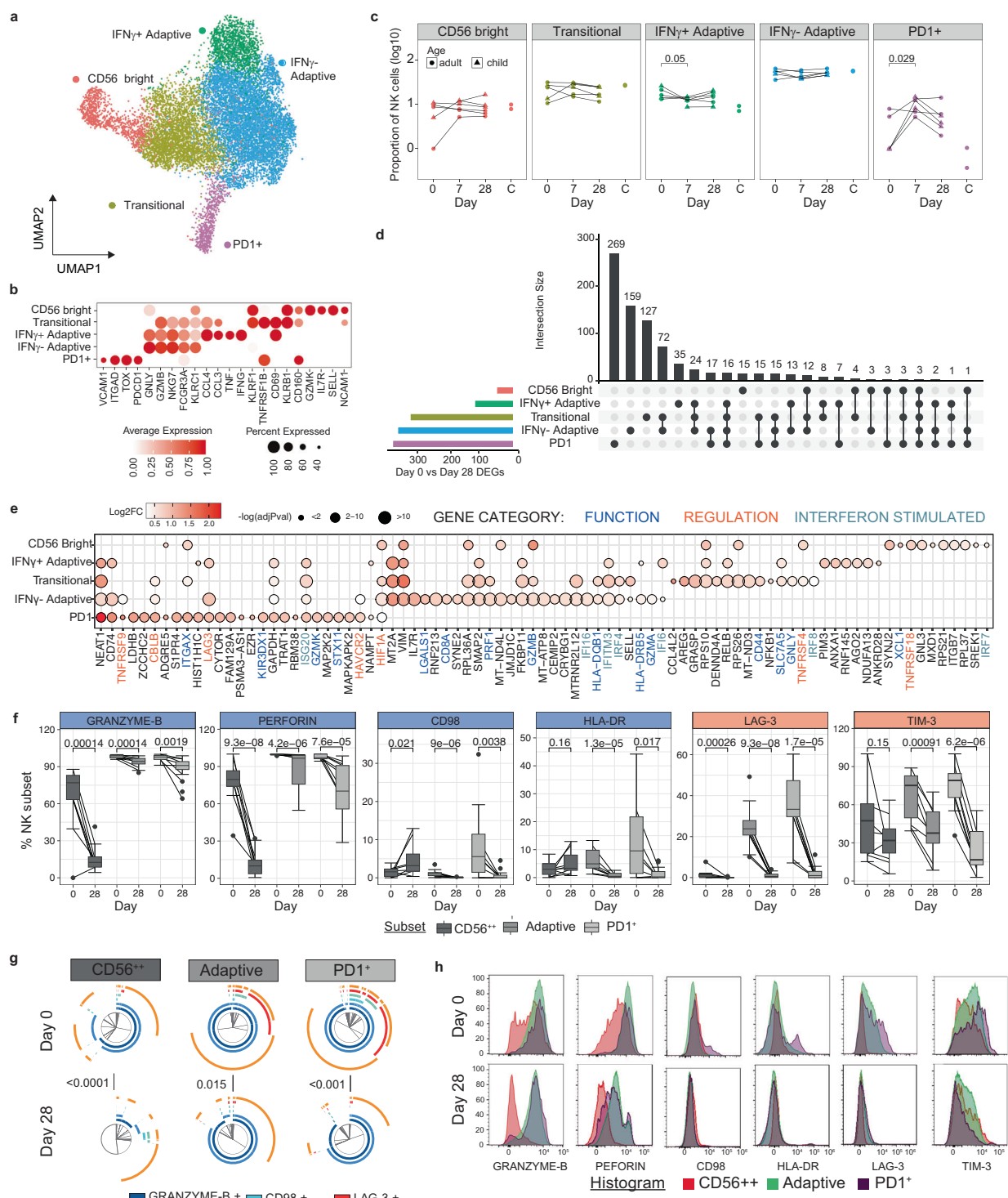

**Fig. 3 | Subset specific activation and regulation of NK cells in malaria. a/b** Five subsets of NK cells were identified based on unsupervised clustering and marker expression as CD56$^{bright}$, Transitional, IFNγ$^+$ Adaptive, IFNγ$^-$ Adaptive and PD1$^+$ subsets. **c** Relative proportion of identified subsets during malaria infection (day 0, $n = 5$), 7- ($n = 6$) and 28-days ($n = 6$) post treatment, and in healthy uninfected individuals ($n = 2$). *P*-value (two-sided) indicated is calculated by Mann-Whitney *U* test between day 0 and indicated subsequent time points. **d** Upset plot of DEGs in NK cell subsets at day 0 compared to day 28. The number of shared and subset specific DEGs indicated. **e** Top 20 DEGs of in each NK subsets, and additional genes of interest with $p ≤ 0.05$. Genes with known roles in regulation and/or function are indicated. Log transformed *p* values (Wilcox test, two-sided, Bonferroni correction). **f** PBMCs from individuals with *P. falciparum* malaria (Day 0, $n = 11$), and

convalescent malaria patients 28-days post-infection (Day 28, $n = 17$) were analysed ex vivo to detect NK cell protein expression of identified genes by flow cytometry. Expression of proteins related to function and regulation of NK cells, shown as positive frequencies of NK cell subsets CD56$^{bright}$ (CD56$^{++}$), Adaptive and PD1$^+$. Box plots show the median and IQR of volunteers, lines represent paired observations, group comparisons performed by Mann-Whitney *U* test (two-sided). **g** Co-expression of proteins related to function and regulation of NK cells analysed by SPICE. Pie graphs comparisons performed by Permutation test. **h** Concatenated histograms normalised to mode show expression of gene-related proteins within NK cell subsets. See also Supplementary Fig. 6-10, Supplementary Table 3, Supplementary Data 3-4. Source data are provided as a Source Data file.

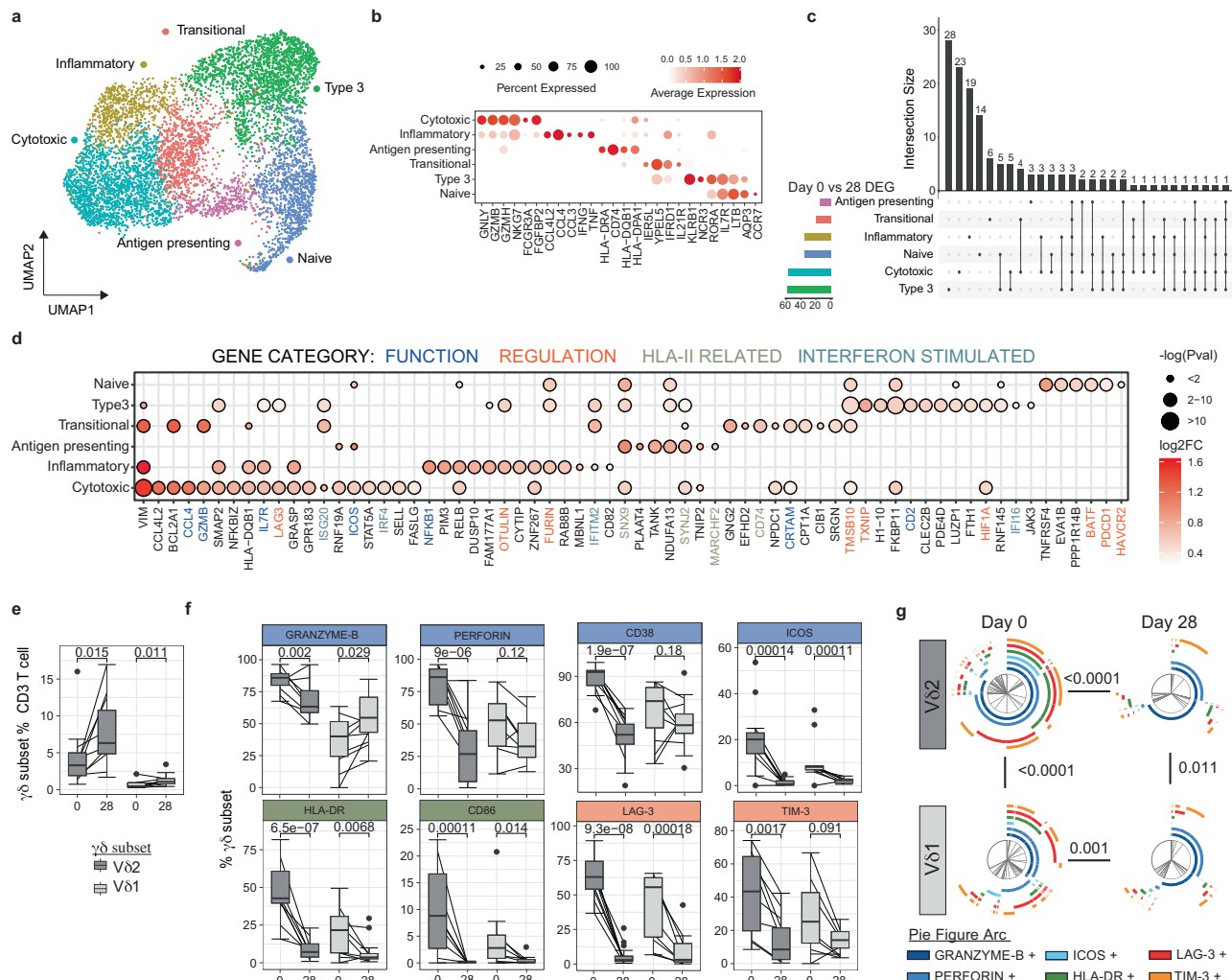

**Fig. 4 | Inflammatory activation and regulation of effector γδ T cells during malaria. a/b** Five subsets of γδ T cells were identified based on unsupervised clustering and marker expression as Cytotoxic, Inflammatory, Antigen-presenting, Transitional, Type 3 and naive. **c** Numbers of upregulated DEGs in γδ T cell subsets day 0 (n = 5) compared to day 28 (n = 6). The number of shared and subset specific DEGs indicated. **d** Top 20 upregulated DEGs in each γδ T cell subsets, and additional genes of interest with p ≤ 0.05. Genes with known roles in regulation and/or function are indicated. Log transformed p values (Wilcox test, two-sided, Bonferroni correction). **e** PBMCs from patients with *P. falciparum* malaria (day 0, n = 11) and convalescent malaria patients 28-days post-infection (day 28, n = 17) were analysed ex vivo to detect Vδ2+ and Vδ1+ γδ T cells and measure protein expression of identified genes by flow cytometry. **f** Expression of proteins related to function and regulation of γδ T cells (day 0, n = 11, day 28 n = 17), shown as positive frequencies of Vδ2+ and Vδ1+ γδ T cells. Box plots show the median and IQR of volunteers, lines represent paired observations, P values (two-sided) indicated are calculated by Mann-Whitney U test. Box and whisker plots indicate first and third quartiles for the hinges, median line, and lowest and highest values no further that 1.5 interquartile range from the hinges for whisker lines. **g** Co-expression of proteins related to function and regulation of Vδ2+ and Vδ1+ γδ T cells analysed in SPICE. Pie graphs comparisons performed by Permutation test (day 0, n = 11, day 28 n = 17). See also Supplementary Fig. 8-12, Supplementary Table 4 and Supplementary Data 5-6. Source data are provided as a Source Data file.

Transitional, Type 3 and Naive γδ T cells (Fig. 4a, Supplementary Data 5). Cytotoxic subset cells expressed the highest levels of *GNLY, GZMB, GZMH, NKG7, FCGR3A* (encoding CD16) and *FGFBP2;* Inflammatory subset cells expressed high levels of *CCL4L2, CCL4, CCL3, IFNG* and *TNF;* Antigen-presenting subset cells expressed HLA-II related genes; and Transitional γδ T cells were characterized by expression of genes that drive the early differentiation of T cells *IL21R, IER5L, YPEL5* and *IFRD1* (Fig. 4b). We also identified Type 3-like γδ T cells with high expression of *KLRB1, NCR3, RORA* and *IL7R,* and Naive γδ T cells with high expression of *LTB, CCR7* and *AQP4,* as in previous scRNAseq data sets[80]. There was an increased proportion of the Cytotoxic cell with the γδ T cells compartment at day 7 and 28 after infection and increased Transitional cell subset at day 7 (p = 0.052, Supplementary Fig. 11A).

To investigate γδ T cell transcriptional changes during malaria, DEGs were identified in γδ T cell subsets, (Fig. 4c, d, Supplementary Data 6). Both shared and subset specific transcriptional changes

between day 0 and day 28 where identified (Fig. 4c, d). Individual contributions to each cell subset and expression of DEGs at day 0 and day 28 is presented in Supplementary Table 4 and Supplementary Fig. 12. DEGs included genes associated with T cell activation (including *VIM, ICOS, IL7R* (encoding CD127)) across multiple subsets, consistent with inflammatory responsiveness during malaria infection[75]. Additionally, consistent with increased cytotoxic capacity during malaria reported previously[76], expression of cytotoxic serine protease *GZMB* was increased in both Cytotoxic and Transitional γδ T cells during malaria. Further, *CRTAM,* which drives the development of cytotoxic CD4 and CD8 T cells[81], was increased during infection, consistent with the expansion of Cytotoxic γδ T cell subsets at days 7 and 28 (Fig. 4d). Along with increased inflammatory and cytotoxic capacity, there was a suggestion of increased antigen presenting properties during infection, with upregulation of HLA-II related genes (*CD74, HLA-DPB1, HLA-DQB1*), and genes related to endocytosis and intracellular

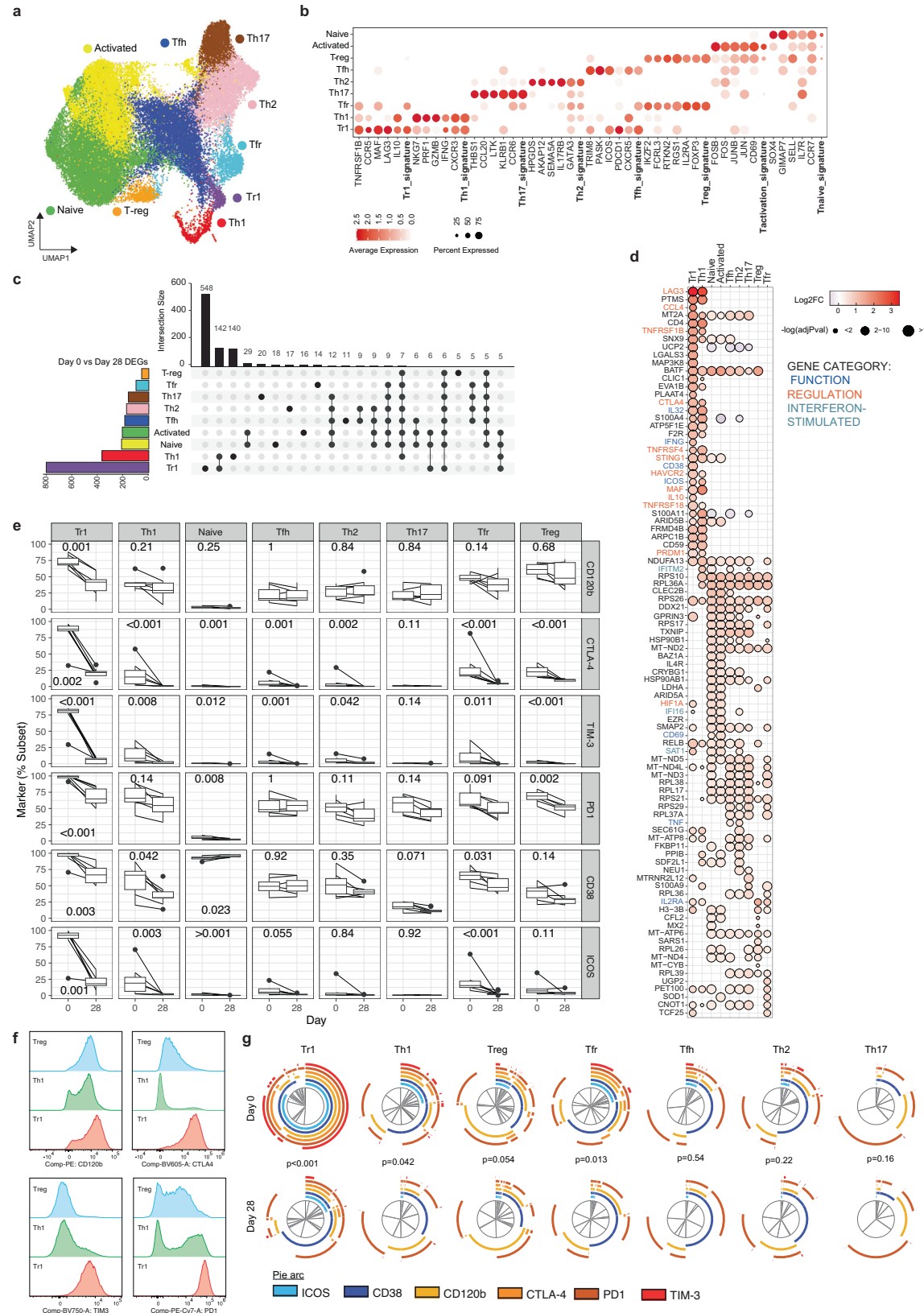

vesicle trafficking (such as *SNX9* and *SYNJ2*). Together, these data are indicative of polyfunctional γδ T cell activation during infection, consistent with previous phenotypic data[75–77].

Along with activation of multiple γδ T cell functions, increased expression of genes with roles in immunoregulation and cell exhaustion were detected (Fig. 4d). These included upregulation of *LAG3* on

Cytotoxic γδ T cells and increased expression of inflammatory regulators *OTULIN*[82] and *FURIN*[83] on Inflammatory γδ T cells. Within Type 3 and Naive γδ T cells, upregulated genes included those related to inflammatory regulation (*TMSB10;* which suppresses inflammatory macrophages)[84], *HIF1A;* which controls γδ T cell mediated inflammation[85] and *BATF;* which is upregulated in exhausted CD8

**Fig. 5 | Tr1 CD4 T cells dominate the response during malaria. a/b** Nine subsets of CD4 T cells were identified based on unsupervised clustering and categorized based on canonical marker expression and T helper expression signatures. **c** DEGs in CD4 T cell subsets at day 0 ($n = 5$) compared to day 28 ($n = 6$) were identified. The number of shared and subset specific DEGs indicated in Upset plot. Overlaps of >5 genes are shown. **d** Top upregulated 20 DEGs in CD4 T cell subsets. Genes with known roles in regulation and/or function are indicated. Log transformed $p$ values (Wilcox test, two-sided, Bonferroni correction). PBMCs from patients with *P. falciparum* malaria (day 0, $n = 7$) and convalescent malaria patients 28-days post-infection (day 28, $n = 9$) were analysed ex vivo to detect CD4 T and measure protein expression of identified genes by flow cytometry. **e** Expression of proteins related to function and regulation, shown as positive frequencies of CD4 T cell subsets: naive (CD45RA$^+$), Tr1 (CD45RA$^-$LAG3$^+$CD49b$^+$), Treg (CD45RA$^-$Tr1$^-$/FOXP3$^+$CD127$^{lo}$),

Tfh (CD45RA$^-$Tr1$^-$/Treg$^-$/CXCR5$^+$), Tfr (CD45RA$^-$Tr1$^-$/Treg$^+$/CXCR5$^+$), Th1 (CD45RA$^-$Tr1$^-$/Treg$^-$/CXCR5$^-$/CXCR3$^+$CCR6$^-$), Th2 (CD45RA$^-$Tr1$^-$/Treg$^-$/CXCR5$^-$/CXCR3$^-$CCR6$^-$) and Th17 (CD45RA$^-$Tr1$^-$/Treg$^-$/CXCR5$^-$/CXCR3$^-$CCR6$^+$). Box plots show the median and IQR of volunteers, lines represent paired observations, $P$ values (two-sided) indicated are calculated by Mann-Whitney $U$ test (day 0 $n = 7$, day 28 $n = 9$). Box and whisker plots indicate first and third quartiles for the hinges, median line, and lowest and highest values no further that 1.5 interquartile range from the hinges for whisker lines. **f** Concatenated histograms normalised to mode show expression of gene-related proteins within CD4 T cell subsets. **g** Co-expression of proteins related to function and regulation of CD4 T cell subsets analysed in SPICE. Pie graphs comparisons performed by Permutation test (day 0 $n = 7$, day 28 $n = 9$). See also Supplementary Fig. 13-17, Supplementary Table 5, and Supplementary Data 7-8. Source data are provided as a Source Data file.

T cells[86]), cell exhaustion (*PDCD1* (encoding PD1); which dampens inflammatory and cytotoxic potential of γδ T cell[87,88], *TNFRSF4* (encoding OX40) and *HAVCR2* (encoding TIM-3); which reduces cytokine and cytotoxic potential of γδ T cells[89] and pro-apoptotic signaling (*TXNIP*). Like myeloid and NK cell responses, DEGs included upregulation of multiple Type I IFN response genes including *ISG20, IRF4, IRF1, IFITM2, IFI16*. γδ T cells have been reported to respond to Type I IFNs produced from poly(I:C) activated cDCs in other settings[90]. However, to the best of our knowledge, a role of Type I IFNs in driving responding γδ T cells during malaria, has not been explored.

To confirm our transcriptional findings, we assessed protein-level expression of multiple functional and regulatory markers in additional patient samples, by flow cytometry (Supplementary Fig. 11B, C). While we were unable to differentiate between Vδ2$^+$ and Vδ1$^+$ γδ T cells in our 3' transcriptional data set, the large majority of circulating γδ T cell within the CD3 T cell compartment at both day 0 during acute infection and day 28 post treatment were Vδ2$^+$ γδ T cells (Fig. 4e). Within Vδ2$^+$ cells γδ T cells, there was increased expression of Granzyme-B and CD38 at day 0, but not in Vδ1+ γδ T cells as detected (Fig. 4f). Other functional and activation markers, Perforin, ICOS, and HLA-DR were increased on both Vδ2$^+$ and Vδ1$^+$ γδ T cells during malaria. Similarly, LAG-3 was increased on both Vδ2$^+$ and Vδ1$^+$ γδ T cells during malaria but TIM-3 was only increased in Vδ2$^+$ γδ T cells (Fig. 4f). Granzyme-B, ICOS and HLA-DR expression was higher on Vδ2$^+$ compared to Vδ1$^+$ γδ T cells during malaria (Supplementary Fig. 11D). Increased parasitemia during peak infection was associated with increased cell frequency and expression of CD38, ICOS and CD86 by Vδ2$^+$ T cells and increased CD86 and HLA-DR by Vδ1$^+$ T cells (Supplementary Fig. 8B). Additionally, Vδ1$^+$ T cell expression of Perforin and TIM-3 increased with age in malaria-infected individuals (Supplementary Fig. 9B). No sex-dependent differences in γδ T cell phenotypes were observed (Supplementary Fig. 10B). To understand the expression of different functional, activation and regulatory markers on Vδ2$^+$ and Vδ1$^+$ γδ T cell subsets, marker expression was analysed by SPICE. Both Vδ2$^+$ and Vδ1$^+$ γδ T cells had high levels of co-expression of key proteins related to multiple functions and regulation (Fig. 4g). The magnitude and composition of marker co-expression by γδ T cells was increased at day 0 compared to day 28 expression.

## CD4 T cell response is dominated by Type 1 regulatory cells during malaria which share signatures with Th1 cells

CD4 T cells play multiple essential roles in protection from malaria, including IFNγ mediated direct killing of parasites, and by providing help to B cells to produce antibodies required for protection[91]. However, multiple lines of evidence have shown that malaria drives the expansion of regulatory CD4 T cells, particularly Type 1 regulatory (Tr1) cells that co-produce IFNγ and IL-10 in this disease. These cells rapidly expand via Type I IFN signalling in initial parasite infection in humans[19], and dominate the parasite specific CD4 T cell response in children in endemic areas[16–18]. To investigate CD4 T cells transcriptionally during malaria, we subclustered CD4 T cells from PBMCs,

with 9 subsets identified (Fig. 5a). These CD4 T cell subsets were annotated as naive, activated, T-regulatory (Treg), T-follicular helper (Tfh), T-follicular regulatory (Tfr), T-helper (Th) 1, Tr1, Th2 and Th17 subsets based on expression of T naive, activation, T-reg and T helper signatures[92–95] and canonical marker genes (Fig. 5b, Supplementary Data 7). The proportions of each scRNAseq identified subset within the CD4 T cells compartment did not significantly change between acute infection and post treatment timepoints (Supplementary Fig. 13). We conducted DEG analysis for each CD4 T cell subset, on day 0 (malaria) compared to day 28 (post treatment (Supplementary Data 8, individual contribution to each subset was consistent, Supplementary Table 5, and DEG day 0 and day 28 expression at the individual level is presented in Supplementary Fig. 14). Tr1 and Th1 CD4 T cells were highly transcriptionally active during infection, while T-regs and Tfr were the least activated (Fig. 5c, Supplementary Data 8). In Tr1 cells, 68% of DEGs were unique, consistent with a Tr1 specific activation program during malaria (Fig. 5c). Upregulated genes included canonical markers of *IFNG* and *IL10*, and many other genes of co-inhibitory receptors with known roles in immunosuppression and/or function of regulatory T cells including *LAG3, HAVCR2* (encoding TIM-3[96]), *TNFR2*[97], *CTLA4*[98], *TNFRSF4* (encoding OX40/CD134[99]), *TNFRSF18* (encoding GITR/CD375[100]), *PDCD1* (encoding PD1) and *CCL4*[101] (Fig. 5d). These upregulated genes were also core signature genes used to identify Tr1 cells within the sub-clustered CD4 T cells at day 0 and 28 (Fig. 5a/b, Supplementary Data 7), suggesting that while the overall frequency of Tr1 cells in the periphery didn't change with infection (Supplementary Fig. 13), the regulatory capacity of Tr1 cells was increased during malaria. The upregulation of multiple co-inhibitory receptors is consistent with our recent data showing Tr1 cells express overlapping co-regulatory proteins[22]. Additionally, *IKZF3* which has been shown to have high expression in IL-10$^+$ CD4 T cells[102], and *LAIR2* which has been identified as a core Treg cell signature gene in humans[103] had increased expression on Tr1 cells during malaria (Fig. 5d). Tr1 cells also showed significant activation during infection, with marked upregulation of *CD38* and *ICOS* during infection. *CD38* upregulation was unique to Tr1 cells, while *ICOS* was also increased on Th1 cells.

Tr1 cells can emerge from Th1 cells that gain *IL10* expression[104]. Consistent with this, most of the top upregulated genes in Tr1 cells were shared with Th1 cells, including regulatory markers *LAG3, TNFRSF4* (encoding OX40), *TNFRSF1B* (encoding TNFR2), *TNFRSF18* (encoding GITR), *HAVCR2* (encoding TIM-3) and *CTLA4* (Fig. 5d). Additional upregulated genes in Th1 cells included *MAF*, which drives *IL10* expression in Th1 cells in murine malaria models[105], *PRDM1* (encoding BLIMP1), which promotes *IL10* in Tr1 cells in murine malaria models[106] and is highly expressed in malaria-specific Tr1 cells in Ugandan children[16], and *STING1* (encoding STING − stimulator of interferon response cGAMP interactor 1), which we have recently shown to be a central driver of Tr1 cell development[107]. In Th1 and/or Tr1 cells, several other Type I IFN responses genes were up regulated, including *IFITM2, IFI16, SAT1, IFI35, IFI27L2, LYE6* and *ISG15* (Fig. 5d,

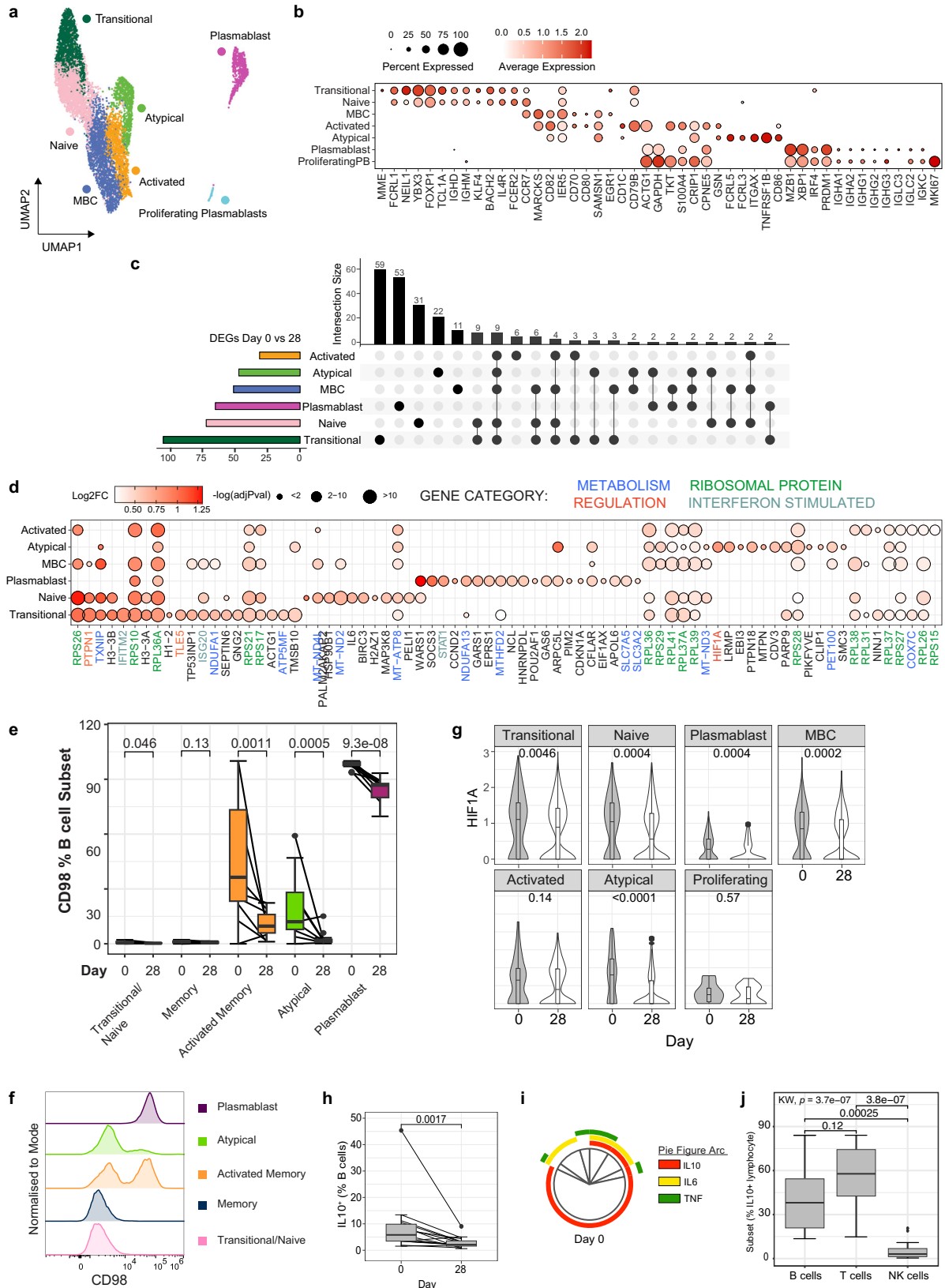

Supplementary Data 8). In other CD4 T cell subsets, the number of DEGs and magnitudes of fold changes to expression were relatively lower, with many ribosomal and mitochondrial genes shared across non-Th1/Tr1 subsets (Fig. 5c, d). However, *BATF*, which is critical for Th17 and Tfh cell differentiation[108,109] was upregulated across all CD4 T cell subsets except for FoxP3⁺ Treg cells. While the roles of Th17 cells in

human malaria are largely unknown, Tfh cell activation and development is critical for the induction of humoral responses required to drive protection against malaria[110]. Tfh cell activation during malaria is skewed towards Th1-Tfh cell responses[111–113], and consistent with this, Th2 and Tfh cells subsets upregulated both the Th1 associated cytokine TNF, and *ETS1* which represses Th2-Tfh subset differentiation in

**Fig. 6 | B cell activation and induction of IL-10⁺ Bregs during infection.**
**a/b** Subsets of B cells were identified based on unsupervised clustering and cate-gorised based on marker expression. **c** DEGs in B cell subsets day 0 (*n* = 5) com-pared to day 28 (*n* = 6) were identified. The number of shared and subset specific DEGs indicated in Upset plot. **d** Top upregulated 20 DEGs in B cell subsets. Genes with known roles in regulation and/or function are indicated, Log transformed *p* values, (Wilcox test, two-sided, Bonferroni correction). **e** CD98 protein expression was quantified on plasmablasts, transitional, atypical, memory and activated memory B cell subsets at day 0 (*n* = 11) and day 28 (*n* = 17). Box plots show the median and IQR of volunteers, lines represent paired observations, group com-parisons performed by Mann-Whitney *U* test (two-sided). **f** CD98 MFI of con-catenated samples at day 0 (*n* = 5). **g** *HIF1A* mRNA expression in each B cell subset at day 0 and 28. *P*-value calculated by Mann-Whitney *U* test *P* (two-sided) is indicated (unadjusted), data from scRNAseq analysis of day 0 *n* = 5 and day 28 *n* = 6 indivi-duals. **h** IL-10 protein expression on B cells during malaria (day 0, *n* = 15) and day 28 post treatment (*n* = 14), Wilcoxon rank sum test (two-sided) is indicated. **i** Co-expression of B cell IL-10 production with IL6 and TNF during malaria (*n* = 15) analysed in SPICE. **j** The proportion of each lymphocyte subset contributing to IL-10 lymphocyte production during malaria (*n* = 15 individuals, at day 0). *P*-value (two-sided) calculated using Kruskal wallis and post-Dunn test (FDR adjusted for mul-tiple comparisons) indicated. Box and whisker plots indicate first and third quar-tiles for the hinges, median line, and lowest and highest values no further that 1.5 interquartile range from the hinges for whisker lines. See also Supplementary Fig. 21-22, Supplementary Table 7 and Supplementary Data 11-12. Source data are provided as a Source Data file.

both human and mouse models of systemic lupus erythematosus[114]. *ETS1* is also essential for *BATF* function in effector T cells[115]. While DEGs of Treg cells and Tfr cells were the lowest of all subsets, *IL2RA*, essential for Treg cell function[116], was upregulated in malaria in both subsets, suggestive of increased functional Treg and Tfr cells during infection. Together these data are indicative of CD4 T cell activation dominated by Tr1 cells with increases suppressive function during malaria. The increase in Type I IFN signalling, is consistent with the the emergence of Tr1 cells from Th1 cells via Type I IFN signalling as shown previously[19,22].

To further investigate the regulatory program in Tr1 and Th1 cells during malaria, we analysed responses on CD4 T cells subset at the protein level in additional study participants at day 0 and 28. CD4 T cells were gated as naive (CD45RA⁺), and non-naive (CD45RA⁻) cells, which were further analysed as Tr1 (LAG3⁺ CD49b⁺), Treg (Tr1⁻/FOXP3⁺CD127ˡᵒ), Tfh (Tr1⁻/Treg⁻/CXCR5⁺), Tfr (Tr1⁻/Treg⁺/ CXCR5⁺), Th1 (Tr1⁻/Treg⁻/CXCR5⁻/CXCR3⁺ CCR6⁻), Th2 (Tr1⁻/Treg⁻/ CXCR5⁻/CXCR3⁻CCR6⁻) and Th17 (Tr1⁻/Treg⁻/CXCR5⁻/CXCR3⁻CCR6⁺) cell subsets (Fig. S15A). Consistent with the emergence of Tr1 cells from Th1 subsets, the majority of Tr1 cells where Th1-like (CXCR3⁺/ CCR6⁻/CXCR5⁻/FoxP3⁻) at day 0 during malaria (median 65.4%, IQR 62.5-73.8%, Supplementary Fig. 15B). Unlike scRNAseq identified subsets, there were increased proportions of Tr1 and Th2 cells, and decreased proportions of naive, Tfh, and Tfr cell subsets at day 0 compared to day 28 (Supplementary Fig. 15C). This difference may be due to the limited surface makers used to identify cells by flow cytometry compared to the overall transcriptional profile used in scRNAseq data, or the stable maintenance of underlying transcrip-tional programs compared to changes of chemokine receptor expression. On each CD4 subset, the proportion of cells expressing regulatory and activation markers CD120b (*TNFRSF1B*), CTLA-4, TIM-3 (*HAVCR2*), PD1 (*PDCD1*), CD38 and ICOS were analysed (Fig. 5e). Consistent with scRNAseq data, at day 0 during malaria, there was significantly higher proportion of Tr1 cells expressing CD120b, CTLA-4, TIM-3, PD1, CD38 and ICOS (Fig. 5e). Additionally, higher proportions of Th1 cells also expressed CTLA-4, TIM-3, CD38 and ICOS during malaria. Increased regulatory and activation mar-kers were also seen on Treg and Tfr cells, but not Tfh, Th2, or Th17 subsets. The expression levels of co-inhibitory markers on Tr1 and Th1 cells was as high or higher than that on FoxP3⁺ Treg cells which are known to have suppressive function (Fig. 5f). Addition-ally, Tr1 cells expressed multiple co-inhibitory receptors during malaria (Fig. 5g). Increased parasitemia during infection was asso-ciated with increased proportions of PD1 expressing Tr1 cells and decreased proportions of CD120b expressing Tfh cells (Supple-mentary Fig. 16A). Age and sex were not associated with regulatory or activation markers on CD4 T cell subsets during malaria (Sup-plementary Fig. 16B and 17). To the best of our knowledge, the high regulatory T cell program seen here within the scRNAseq malaria data set has not been reported in other diseases[38–40,117]. Together

data is supportive of specific upregulation of a regulatory program within CD4 T cells during malaria.

The role of CD8 T cells in immunity to *P. falciparum* blood stage malaria is unclear, particularly due to the lack of MHCI on the surface of RBCs. However, limited studies have reported activation and increased function of CD8 T cells during *P. falciparum* malaria, parti-cularly in hospitalised patients[118]. To investigate CD8 T cell activation in acute malaria, CD8 T cells identified in PBMCs were subclustered. Naive/Central memory, Memory progenitor, Effector memory and Cytotoxic effector cells were identified (Supplementary Fig. 18A, B, Supplementary Data 9), the proportion of which was largely consistent between infected (day 0) and convalescence (day 28) (Supplementary Fig. 18C). We conducted DEG analysis for each CD8 T cell subset, on day 0 (malaria) compared to day 28 (post treatment (Supplementary Data 10, individual contribution to each subset was consistent (Sup-plementary Table 6), and DEGs at the individual level is presented in Supplementary Fig. 19). Compared to CD4 T cells only a small number of DEGs between day 0 and 28 were identified, the largest number being in cytotoxic effector CD8 cells (Supplementary Fig. 18D, Sup-plementary Data 10). Upregulated genes included those relating to increased inflammation and cytotoxicity (*IFNG, TNF, CCL4, CCL3, IFNGR1, GZMB, PRF1*) (Supplementary Fig. 18E). Genes associated with regulation were also upregulated in Effector subsets but no other CD8 T cell subsets (including *LAG3* and *CTLA4)* (Supplementary Fig. 18E). While CD8 T cells can also differentiate into a phenotypically distinct regulatory lineage and produce IL-10 like Tr1 CD4 T cells in other diseases[119], DEGs during malaria in CD8 T cells subsets had limited overlap with DEGs that dominated the Tr1/Th1 response in CD4 T cells, and no evidence of increased *IL10* expression was detected (Supple-mentary Fig. 18F, G).

### Expansion of IL-10⁺ regulatory B cells during malaria infection

Malaria specific B cell responses are essential for development of immunity against malaria, with antibodies being key mediators of protection through control of parasite burden[120]. While robust mem-ory B cells and sustained antibodies can develop against malaria[25], there is evidence that these responses may also be negatively impacted by malaria driven immunomodulation. Memory B cell responses are suboptimal in some malaria transmission settings, with lower levels of antibody production, short-lived antibody responses and expanded atypical memory B cells[23–25]. Atypical memory B cells express high levels of FCLR5 and appear to have reduced functional capacity com-pared to 'typical' memory B cells[26,27]. However, atypical responses emerge in both infection and vaccination[121], mount recall responses[122] and can produce antibodies with the support of T-follicular helper cells[123]. Therefore, whether atypical memory B cells are protective or disruptive in protective immunity remains unclear.

To investigate transcriptional changes to B cells during malaria within our PBMC data set, B cells were sub-clustered to identify 7 subsets of B cells (Fig. 6a). Based on previously published

studies[121,124,125], identified subsets included transitional B cells with high expression of *MME (*encoding CD10) and naive B cells with high expression of *TCL1A, NEIL1, IGHD, IGHM, FOXP1, BACH2* and *IL4R;* quiescent memory B cells which had relatively high expression of memory marker *CCR7*, and upregulated *MARCKS, CD82, IER5, CD70 and CD80* which are increased in expression on memory relative to naive B cells in previously published data sets[126,127] annotated by the Human Protein Atlas (proteinatlass.org); activated B cells with increased expression of *CD1C, CD79B, ACTG1, GAPDH, TKT, S100A10, CRIP2, CPNE5,* and *GSN;* Atypical memory B cells with high expression of *FCLR5, FCLR3, ITGAX* (encoding CD11c)*, TNFRSF1B* and *CD86;* plasmablasts with expression of *MZB1, XBP1, IRF4, PRDM1 (*encoding BLIMP1), and switched IgG genes; and proliferating plasmablasts which expressed plasmablast marker genes along with high levels of *MKI67* (Fig. 6b, Supplementary Data 11). During infection (day 0), there was an increased proportion of plasmablasts, which made up to 20% of the B cell compartment, consistent with previous studies of in Ugandan[128] and Kenyan children with malaria[129], and in adults during controlled human malaria infection[130] (Supplementary Fig. 20A).

DEG analysis of each B cell subset comparing day 0 to day 28, identified large numbers of DEGs across all subsets except proliferating plasmablasts (Fig. 6c, Supplementary Data 12, individual cell numbers Supplementary Table 7, and DEG expression at day 0 and day 28, Supplementary Fig. 21). DEGs were both shared and subset specific, and a large number of the top DEGs for each subset were ribosomal proteins, possibly indicating increased protein synthesis and highly activated states of B cells during infection (Fig. 6d). Additionally, many DEGs had roles in metabolism, consistent with reshaping of energy use and metabolic programs during B cell activation[131] (metabolic associated DEGs include members of NADH dehydrogenase complex *NDUFA1, NDUFA13, MT-ND4L, MT-ND2, MT-ND3*, cytochrome C oxidase subunit *COX7C* and chaperone *PET100*, ATP synthase subunits *ATP5MF* and *MT-ATP8*). Across B cell subsets, multiple upregulated Type I IFN signaling response genes were detected, including *IFNITM2, ISG20*, and *STAT1*, consistent with the important role of Type I IFN signaling in malaria immune responses across multiple cell subsets. There was evidence that metabolic remodeling during infection was B cell subset specific. For example, transitional, naive, memory B cells and atypical memory B cells during acute infection had increased expression of *TXNIP*, a glucose feedback sensor which inhibits glucose uptake[132,133]. In contrast, plasmablasts had increased *SLC7A5* (encoding LAT1) and *SLC3A2* (encoding CD98 which interacts with LAT1 to transport L-glutamine), consistent with our recent findings of the importance of plasmablasts as negative regulators of germinal centre development via acting as a nutrient sink in mice models of malaria[130]. To confirm upregulation of glutamine transport on plasmablasts, CD98 levels were measured across B cell subsets in additional patients. We assessed CD98 expression during and after malaria in plasmablasts (CD27+CD38+), transitional/naive IgD+ B cells, and IgD- B cell subsets (atypical [CD27-CD21-], memory [CD27+CD21+] and activated memory [CD27+CD21-]) (Supplementary Fig. 20B, C). In these additional participants, plasmablasts made up to 30% of the B cell compartment during malaria, but less than 2% at 28 days post-treatment (Supplementary Fig. 20D). There was also a significant increase in the proportion of activated memory cells during malaria (Supplementary Fig. 20D). The frequency of CD98+ cells in each subset increased during malaria in plasmablasts, transitional/naive, atypical and activated memory B cells (Fig. 6e). However, the frequency of CD98+ cells, and the magnitude of CD98 expression was far greater on plasmablasts compared to other subsets (Fig. 6f, Supplementary Fig. 20E, F). Together, these data are consistent with a potential negative role of plasmablast expansion and CD98 expression as a nutrient sink that limits productive germinal center activation during infection[130], but shows that activated memory B cells also upregulate glutamine transport during infection.

Along with a potential disruptive role of plasmablasts in malaria infection, several upregulated DEGs where suggestive of other tolerized/immunosuppressed B cell responses during infection. For example, *PTPNI* (encoding PTP1B) which was upregulated in transitional, naive and memory B cell subsets, negatively regulates B cell signally via CD40, BAFF-R and TLR4, and downregulates T-dependent immune responses[134]. *TLE5* (also known as *AES*), which negatively regulates NF-κβ signaling[135], required for B cell activation and survival[136], was upregulated on transitional B cells during infection. Of note, *HIF1A*, which drives B cell *IL10* production in hypoxic conditions[137], was significantly upregulated in Atypical B cell during acute infection. Further interrogation of *HIF1A* suggested increased expression during acute infection also occurred in transitional, naive, memory B cells and plasmablasts subsets (Fig. 6g), consistent with the capacity of diversity of human B cells subsets to produce IL-10[138]. IL-10 production by B cells is indicative of Breg subsets, which have been shown in mice to be a major source of IL-10 during experimental malaria infection, and protect from experimental cerebral malaria[28,139], however have yet to be identified during human malaria. To confirm the expansion of IL-10+ Bregs during *P. falciparum* malaria, we quantified IL-10+ production in B cells in additional study participants. We analyzed the total B cell population, due to the diverse B cell phenotypes of IL-10+ Bregs, and measured TNF and IL-6 production, which are often co-produced with IL-10[138] (Supplementary Fig. 22A). Data showed there was significant increase in IL-10 production in B cells during malaria, indicating of malaria induction of Bregs (Fig. 6h). In contrast, there was no evidence for increased B cells expression of IL6 nor TNF, despite increased *IL6* transcript levels (Supplementary Fig. 22B). IL-10 B cell production was not associated with parasitemia, age or sex (Supplementary Fig. 22C/D/E). Although a previous study reported co-expression of TNF and IL6 by IL-10 producing Bregs[138], only a small fraction of IL-10+ Bregs co-produced IL6 during malaria, and there was minimal co-expression with TNF (Fig. 6i). To assess the relative importance of Bregs, compared to IL-10+ T cells (largely Tr1 CD4 T cells), the proportion of B cells amongst all IL-10 producing lymphocytes was measured. The proportion of IL-10 lymphocytes that were Bregs was comparable to CD3 cells as the source of IL-10 from lymphocytes (Fig. 6j), identifying IL-10 Bregs as a potentially important contributor to the regulatory/tolerogenic response during malaria infection in humans.

## Discussion

Malaria drives tolerogenic immune cell responses which protect from inflammation mediated immunopathogenesis at the costs of reduced parasite control and suboptimal adaptive immunity. Here, using scRNAseq analysis of PBMCs during and following *P.falciparum* malaria, we comprehensively map malaria associated tolerogenic responses across innate and adaptive immune cells. These data show that malaria driven immunomodulation occurs across the immune landscape, with subset specific activation and regulatory programs identified. By analysing malaria-driven transcriptional changes at the subset level, we not only increase our understanding of how malaria modulates specific immune cell subsets, but also identify IL-10+ Bregs as a major tolerogenic response in human adaptive immune cells during infection.

The use of scRNAseq analysis PBMC immune responses during other infections has identified key protective and disrupted responses in various disease states[39,117,140,141]. Here, we leveraged a large data set of >40,000 cells to understand malaria driven immune responses within major immune subsets, but also within subclustered cells. This approach allows for a granularity of understanding of subset specific cell responses not previously possible with bulk-level analysis. Within innate myeloid cells, malaria drove changes to monocytes consistent with the induction of immunosuppressive MS1-like monocytes, which have high expression of alarmins *S100A8/A9*, along with *RETN* and *ALOX5AP* and reduced expression of MHC class II. These

immunosuppressive monocytes have been identified in scRNAseq data sets in both sepsis and COVID-19 patients (particularly those with severe disease), but not HIV infected individuals or during dengue infection[38–40,117]. During sepsis and COVID-19, immunosuppressive monocytes appear to emerge directly from inflammation-induced myelopoiesis within the bone marrow[38,142]. This pathway may also be important in malaria, with parasite infection shown to drive emergency myelopoiesis in mouse models[143]. How these immunosuppressive monocytes protect from parasite-mediated immunopathology is unknown, however, the importance of tolerized monocytes in anti-disease immunity to malaria has been suggested by others[11]. This anti-disease protection may come at a cost to both adaptive immunity by disruption of antigen presentation via down regulation of HLA-DR, which was also seen in DCs[8], and more broadly. For example, in sepsis, immunosuppressive monocytes have reduced responsiveness to LPS (TLR4 stimulation), consistent with dysregulated response to future bacterial infection in these patients[38]. Reduced responsiveness to LPS has also been reported for monocytes exposed to *P. falciparum* parasites in vitro[11]. As such, immunosuppressive monocytes may also be an important factor in the increased risk of bacterial infection in children with recent or acute malaria[144]. Despite the down regulation of cytokines and chemokines within the myeloid compartment, we also observed that many receptors involved in antibody mediated parasite control, including *CD163, CD93, FCGR1A & 2A, FCAR* and *CR1* were upregulated, consistent with recent reports by others[48], and an important role in monocytes in antibody Fc mediated protection. As such, while inflammatory monocyte responses are diminished, other functions are maintained during acute infection.

Evidence of malaria induced immunosuppression within our data was also observed in NK and γδ T cell subsets, where multiple co-inhibitory receptors were upregulated during infection (including *TNFRSF4, TNFRSF9, TNFRSH18, HAVCR2, LAG3,* and *PDCD1*). Co-inhibitory receptors play important roles in regulating immune response to chronic infections, including malaria[145]. For NK cell responses, increased PD1 expression has been previously reported in malaria exposed individuals previously[13], however the roles of other co-inhibitory receptors in regulating NK cell responses during malaria have not been investigated. With others, we have recently shown that a CD56[neg] NK cell subset is expanded in areas of high malaria burden and have important functional roles in protecting from diseases via antibody dependent cellular cytotoxicity[146]. These CD56[neg]NK cells had high expression of LAG-3, a molecule which has been shown in other studies to be expressed on NK cells with increased glycolytic activity and to negatively regulate NK cytokine production but not cytotoxic activity[71]. Here, LAG-3 and other co-inhibitory receptors appeared to be expressed to the highest levels on PD1[+] NK cells, which contained both CD56[++/bright] and CD56[dim] cells. PD1[+] NK cells were the most highly activated NK cells during malaria, and had high granzyme-B and perforin expression, consistent with retained cytolytic capacity. Further studies are required to understand the relationship between CD56[neg] NK subsets, and PD1[+] cells.

Within γδ T cells, upregulation of co-inhibitory receptors on Vδ2[+] cells to mediate tolerance has previously been shown in areas of high malaria burden[14]. Indeed, several of these regulatory genes (*BATF, HAVCR2, TXNIP*) have been previously shown to be increased transcriptionally and at the protein level in Vδ2[+] γδ T cells in children with recent or high levels of repeated infection[14,78,89]. While Vδ1[+] γδ T cells are less well studied in malaria, some LAG-3 expression was detected on Vδ1[+] γδ T cells during controlled human malaria infection of Tanzanian adults[147]. Here we show that multiple co-inhibitory receptors are also upregulated on γδ T cell during an acute infection in a low transmission setting, on both Vδ2[+] cells, which dominated the response, but also on Vδ1[+] γδ T cells. In areas of high malaria burden, Vδ2[+] γδ T cells with inhibitory receptors maintain or have enhanced cytolytic capacity and antibody dependent functions[79], and recent

studies have suggested that on Vδ1[+] γδ T cells with increased cytotoxic capacity expand with repeated malaria infection[148]. Consistent with this, co-inhibitory markers were upregulated concurrently with increased Granzyme-B and perforin, indicative of cytolytic function. Together, these data suggest that regulatory proteins have the potential to play a role in controlling inflammation, while other cytotoxic functions of NK and γδ T cells are maintained.

Within CD4 and B cell subsets, tolerogenic responses appeared dominated by a major upregulation of *IL10*. Within CD4 T cells, the Tr1 cell subset was the most highly activated during malaria, and these cells had significantly increased transcription of both canonical cytokines *IL10* and *IFNγ*, and multiple co-inhibitory receptors (including *LAG3, OX40, TNFR2, GITR, TIM3* and *CTLA4*). A large proportion of malaria driven DEGs in Tr1 cells were shared with Th1 cells, consistent with an emergence of Tr1 cells from the Th1 cell subset[104]. Indeed, the majority of Tr1 cells identified at the protein level expressed Th1 chemokine receptor CXCR3. DEGs in Tr1 cells also included transcriptional factors with known roles in Tr1 cell development (*MAF, PRDM1* and *STING*)[22,105–107]. Similarly, within B cell subsets, malaria drove a significant increase in *HIF1A* expression, which has previously been shown to be a critical transcription factor for the induction of IL-10 producing Bregs in mouse models[137]. Consistent with this, we show significantly increased ex vivo secretion of IL-10 from B cells during malaria compared with 28 days post-treatment in additional individuals. Indeed, during infection B cells were a major contributor of IL-10 from lymphocytes during malaria. While IL-10 producing CD4 T cells have been well recognised in malaria[16,18,22,106,107,149], this study identifies B cells as a major source of IL-10 during human malaria. Further studies are needed to understand the development of Bregs during malaria, their roles in anti-disease and anti-parasitic immunity and/or immunosuppression[150]. Additionally, the potential link between Tr1 cells as the driver of Breg activation to suppress inflammation and disease as shown in other settings should be investigated[151].

Taken together, data here and previous studies, highlight the multiple tolerogenic responses induced during malaria across the immune landscape. These regulatory landscapes are not common across all infections, and the upregulation of multiple co-inhibitory markers have not been detected in comparable scRNAseq analysis of PBMCs during SARS-CoV-2, influenza, HIV or dengue[38–40,117]. A potential link of malaria-induced tolerogenic responses across all immune cell subsets, is Type I IFN signalling, with Type I IFN responses and increased transcription of IFN-stimulated genes detected across the immune landscape. While first described in viral infection as critical effector cytokines, other studies have also shown that Type I IFNs exhibit immunoregulatory effects that impeded control of some non-viral pathogens[152], including protozoan parasites[153]. In malaria infection, Type I IFNs have both protective and detrimental impacts on immune response, parasite clearance and protection from immuno-pathogenesis, dependent on timing, parasite species and model system (reviewed in ref. 29). Indeed, previous studies in human experimental infection have shown that *P. falciparum* parasites rapidly induce Type I IFNs that drive development of regulator Tr1 CD4 T cells[19]. While not directly investigated in malaria, Type I IFN signalling has been reported to negatively regulate NK cell inflammatory response during viral infection[74], promote upregulation of LAG-3 on NK cells in healthy donors[71], and drives Breg induction in helminth infection[154]. While the causative role of Type I IFN signalling in driving the upregulation of regulatory cell phenotypes seen here remains to be confirmed mechanistically, our team is currently exploring whether host-directed therapies that transiently block Type I IFNs may have therapeutic roles in enhancing protective anti-parasitic immunity[155,156].

A number of limitations of our study should be noted. The primary aim of this study was to identify malaria driven transcriptional changes at the cell subset level. However, once PBMC cells are subclustered, relatively low cell numbers were available for analysis to

identify malaria DEGs in rare cell types. As such, we were not able to identify DEGs at the individual donor level, and therefore may have overlooked individual level heterogeneity. While the overall expression levels of DEGs at day 0 and 28 were consistent across individual, we further investigated this potential heterogeneity by confirming key transcriptional changes at the protein level in additional patients. In all cases, we considered day 28 'convalescent' time point, but it should be noted that others have reported sustained changes to multiple cell subsets up to a year after a malaria infection[48], and there were significant DEGs between day 28 and endemic control cells, as such day 28 may not be a completely 'healthy' time point. Future studies are required to better understand maintained changes at day 28 post infection, along with more detailed study of immune cell activation between treatment (day 0) and day 7. The limited number of individuals analysed by scRNAseq also precludes the analysis of the impact of other host factors such as age, sex, parasite burden and/or disease manifestations on transcriptional changes. Future studies could take advantage of rapidly developing technologies to increase cell numbers and/or individuals used in transcriptional studies. Additional technical limitations include the use of 3' sequencing, as such clonal development and TCR/VDJ usage was not investigated. Additionally, only PBMCs responses were assessed, and how these peripheral responses related to the immune response within tissues is unknown. Further, malaria antigen specific T and B cells were not assessed, and future studies are required to specifically investigate antigen specific response during and after infection. Our study only investigates one study site with all patients >3 years of age and presenting with uncomplicated malaria where other important genetic factors are unknown (such as sickle cell trait which influences malaria symptoms and immune responses[157]), and as such the broad application of results to other malaria transmission settings, young patients, known genetic backgrounds and/or disease phenotypes is unknown. Finally, while data is indicative of a tolerogenic response across multiple immune cell populations, further mechanistic studies are required to confirm that these cell responses are functionally suppressive. Future studies are also required to investigate whether Tr1/Th1 CD4 cells and Bregs induced in malaria responding in an antigen specific, or globally suppressive manner.

In conclusion, we use scRNAseq analysis of a large number of PBMC cells to make a granular level study of immunoregulatory responses across the immune landscape during *P. falciparum* malaria. All data sets and interactive integrated scRNAseq data file is made publicly available for future analysis of the malaria immune landscape by the research community.

## Methods

### Study design
To investigate malaria driven transcriptional changes in specific immune cell subsets, we sorted live PBMCs from 6 malaria infected donors (day 0), and two subsequent time points after drug treatment (day 7 and 28), along with PBMCs from 2 healthy controls. We performed scRNAseq of these cells and used clustering and sub-clustering to identify specific immune cell subsets. Differential gene analysis between day 0 and day 28 was performed for each cell cluster and sub-cluster. Key transcriptional changes were confirmed at the protein level with additional donor samples. Patient demographics are in Supplementary Table 1.

### Ethics statement
Ethics approval for the use of human samples was obtained from the QIMR Berghofer Human Research Ethics Committee (HREC P3444), the Northern Territory Department of Health and Menzies School of Health Research ethics committee (HREC 2010-1431), and Medical Research and Ethics Committee, Ministry of Health, Malaysia (NMRR-10-754-6684 and NMRR-12-499-1203). Written informed consent was obtained from all adult study participant or, in the case of children, parents or legal guardians. Participants were compensated for transport costs for follow up visits to clinics following discharge.

### Study participants and peripheral blood mononuclear cell processing
Peripheral blood mononuclear cells (PBMCs) and plasma were obtained by convenience sampling of patients with uncomplicated malaria from previous conducted clinical trials and cohort studies of malaria conducted in Sabah Malaysia between 2010 and 2016. During this period, *P. falciparum* malaria was low in Sabah Malaysia, with an estimated incidence of 0.18/1000 people in 2011[158]. These studies included a randomized controlled trial of artemether-lumefantrine against chloroquine[159], and a clinical efficacy study of oral artesunate followed by oral mefloquine[160], both conduced on patients with acute malaria at three district hospital sites (Kudat, Kota Marudu and Pitas)[160,161], and a prospective observational study which malaria patients conducted at the tertiary referral center (Queen Elizabeth Hospital, Kota Kinabalu)[162]. Within these studies, non-pregnant patients (aged >3 years) with microscopy-diagnosed malaria who provided written informed consent were enrolled. For the drug treatment trials, treatment for individuals followed randomization. All individuals regardless of treatment were aparasitemic by 72 hours, confirmed by thick smear or PCR[159,160]. For the observational study, treatment followed hospital guidelines: artemether-lumefantrine for *P. falciparum* and *P. knowlesi* malaria; chloroquine and primaquine or ACT for *P. vivax* malaria; and intravenous artesunate for severe malaria, or if deemed warranted by the treating clinician. Severe malaria was defined according to 2014 WHO guidelines, and patients with severe disease were not included in current immunology study. No treatment failures were reported. Across all trials/cohorts a higher proportion of infected males was observed across all malaria species, possibly because of infection risk of forest worker[161]. Within these parent studies, blood samples were collected and archived for future studies from individuals at enrolment (acute malaria timepoint, day 0), and 7- and 28-days post-treatment. Additionally, samples were collected from individuals who had been living in the area in the preceding 3 weeks, who were negative for *Plasmodium* by microscopy, and who had no history of fever in previous 48 hours were as endemic healthy control samples (*EC*). Blood was collected in lithium-heparin collection tubes, then PBMCs and plasma were isolated from whole blood via density centrifugation with Ficoll-Paque prior to cryopreservation.

For the current immunology study, individuals for scRNAseq analysis were selected from parent studies based on sample availability of repeated blood samples at day 0, 7 and 28 after enrolment/treatment and confirmed uncomplicated *P. falciparum* mono-infection by PCR. For phenotypic analysis, further individuals from the same parent clinical trials/cohort were selected based on sample availability at day 0 (acute malaria and enrolment) with uncomplicated malaria and confirmed *P. falciparum* mono-infection by PCR. Not all day 0 samples also had a day 28 sample available, so additional day 28 (convalescence) time points were selected. Sex (self-determined) and age, reported in Supplementary Table 1, were not considered in sample selection, but all sex and age associations are reported in supplementary figures. Patient had a range of parasitemia during infection and clinical characteristics, which are presented in Supplementary Table 1. Cytomegalovirus sero-status was measured using plasma samples using the Human Anti-cytomegalovirus IgG ELISA Kit (CMV) ab108724. 100% of individuals were CMV positive (Supplementary Table 1).

### 10X Genomics Chromium GEX Library preparation and sequencing
PBMC samples were thawed in RPMI 1640 (Gibco) containing 10% FCS and 0.02% Benzonase. 1E6 PBMCs were stained for viability with

Propidium iodide (PI), and live cells were sorted on BD FACSAria III Cell Sorter into 2% FBS/PBS and counted on hemocytometer. Up to 10,000 cells were loaded into each lane of Chromium Next GEM Single Cell 3′ Reagent Kit v3.1 and Gel Bead-in-Emulsion (GEMs) generated in Chromium Controller (Chromium Next GEM Single Cell 3′ GEM, Library & Bead Kit v3.1, PN-1000121, Chromium Next GEM Chip G Single Cell Kit PN-1000120, Single Index Kit T Set A, PN-1000213). Samples were run in two batches, which were later integrated. 3′ Gene Expression Libraries were then generated according to manufacturer's instructions. Generated libraries were sequenced in a NextSeq 550 System using High Output Kit (150 Cycles) version 1 according to manufacturer's protocol using paired-end sequencing (150-bp Read 1 and 150 bp Read 2) with the following parameters Read 1: 28 cycles, Index 1: 8 cycles, Read 2: 91 cycles.

### scRNAseq transcriptomic analysis

**Pre-processing of raw sequencing files.** Single cell sequencing data was demultiplexed, aligned and quantified using Cell Ranger version 3.1.0 software (10x Genomics) against the human reference genome (GRCh38-3.0.0), with default parameters. Raw sequencing data and processed Cell Ranger outputs are found at GSE217930, https://www.ncbi.nlm.nih.gov/geo/query/acc.cgi?acc=GSE217930. Cell ranger count matrices for each sample (donor and day) were loaded, merged and analysed using Seurat package v4[163].Cell cycle scores, mitochondria DNA transcripts and complexity score (log10genes per UMI) were calculated per cell using Seurat built-in functions. Cell cycle score was assigned to each cell using the CellCycleScoring function and evaluated with Principal Component Analysis (PCA). Cells with less than 20% of mitochondria DNA transcripts, complexity score higher than 0.8 and at least 250 genes and 500 UMIs were retained (Supplementary Fig. 1A). We observed a low cell number and mitochondrial ratio in the 'child1day0' sample due to a 10X Chromium wetting error, therefore this sample was removed from further analysis (Supplementary Fig. 1B). At the gene level, any haemoglobin-associated genes and genes expressed in less than ten cells were filtered out. The filtered dataset was scaled to regress out the effects of mitochondria DNA transcripts content and cell cycle using the ScaleData function, which regresses each variable individually. We followed the integration workflow included in Seurat to remove sources of variation due to donor (Supplementary Fig. 1C). To do so, filtered data was split per donor and day. Each dataset was normalise using the NormalizeData function, and the most variable genes in each of them were selected using FindVariableFeatures. Before integration, the most variable genes shared among the datasets were identified using FindIntegrationAnchors and used to integrate the datasets using IntegrateData function. During integration we also performed CellCycleScoring to improve the accuracy and robustness of downstream analyses by accounting for the effects of the cell cycle on gene expression.

**Cell clustering and sub-clustering.** We applied a linear transformation using ScaleData and regressed out mitochondrial contamination a source of unwanted variation. The top 30 principal components (PCs) were calculated using RunPCA. From these 30 PCs we constructed a k-nearest neighbour graph Euclidean distance in PCA space and refine the edge weights between any two cells based on the shared overlap in their local neighbourhoods (Jaccard similarity), based on standard Seurat workflow. Next, we applied the Louvain algorithm as our modularity optimization technique to set the granularity of clusters at a resolution of 0.6. The non-linear dimensionality reduction technique 'UMAP' was used to visualise cluster in 2D space. Clusters were annotated based on canonical marker expression, with 15 cell subsets identified (Supplementary Fig. 1C). NK, γδ T, CD4 T, CD8 T and B cells clusters were extracted

into individual Seurat objects after scaling, we identified principal components, clustered with the Louvain algorithm at unique resolutions for each subset and annotated by dimensionality and expression of genes related to canonical subset phenotypes as above. For all cell subsets, integration allowed for identification of cell subset clusters across all individuals (Supplementary Fig. 23).

For CD4 T cells, T helper signatures from prior publications; Tfh[92], Tr1[95] and Th1, Th2, Th17, Treg[93] were analysed within sub clusters. For high level cell clusters, and sub-clustering analysis, marker genes for all clusters (outputs from 'FindAllMarkers') are in Supplementary Data 1, 3, 5, 7, 9, 11. Individual contributions to identified clusters, and total cell numbers in each cluster are found in Supplementary Tables 2-7. Processed Seurat file "annotated_Sabah_data_21Oct2022.rds" is found at https://doi.org/10.5281/zenodo.6973241. Analysis code can be accessed at https://github.com/MichelleBoyle/scRNAseq_malaria_2023.

**Differential gene expression analysis.** To identify genes differentially expressed during malaria, 'FindMarkers' function with default parameters in Seurat (including *p* value calculated with Wilcox test, two sided, with Bonferroni correction) was used comparing specific cell subsets at different time points. DEGs for each cluster and subcluster are found in Supplementary Data 2, 4, 6, 8, 10, 12. Identified DEGs were analysed via Gene Set Enrichment Analysis, using GSEA publicly available software (https://www.gsea-msigdb.org/gsea/index.jsp)[164,165], to identify significantly enriched gene ontology (GO) terms for each high-level cluster and subcluster during malaria infection. The HOMER v4.9 package was used to identify significantly overrepresented upstream regulators, of cis-elements 1 kb upstream of the transcription start site (TSS) of the DEGs using the findMotifs.pl script[166]. Shared and subset specific DEGs were visualised using Upset plots using UpSetR package v1.4.0[167]. DEGs were identified as a tolerogenic signature based on previously described functions as co-inhibitory receptors[145].

### Flow cytometric cell phenotyping comparison of scRNAseq samples

2 million cells from the same PBMC vial as used for scRNAseq was used for phenotyping of major cell subsets (Supplementary Table 1). Cells were stained at room temperature with LIVE/DEAD Fixable Blue and surface markers with antibodies purchased from BD Biosciences or Biolegend (Supplementary Table 8). Data were acquired with 3-laser Cytek Aurora, and subsets identified as described in Supplementary Fig. 1.

### Flow cytometric ex vivo cytokine production analysis

PBMCs were thawed and 1 million cells incubated at 37°, 5% CO₂ for 2 hours without additional stimulation as above, in additional study patients with *P. falciparum* malaria at day 0 and day 28 post-treatment. Protein transport inhibitor containing Monensin and protein transport inhibitor containing Brefeldin A were added to cells (both 10 μg/ml, BD Biosciences) and cells cultured for an additional 4 hours to capture the cytokine production of these cells due to current/recent infection without further stimulation ex vivo via re-stimulation. Cell surface staining was performed at RT for 15 minutes with a panel of antibodies purchased from Biolegend, BD Biosciences or Miltenyi-Biotec (Supplementary Table 9). Following 2 washes with 2%FCS/PBS, cells were permeabilised with BD cytofix/cytoperm solution for 20 minutes on ice. Intracellular staining (ICS) was performed following this to assess cytokine production. Intracellular staining was performed using antibodies listed in (Supplementary Table 9). Samples were incubated with the antibodies for 30 minutes on ice, washed twice with BD perm wash buffer then fixed with BD stabilising fixative. All samples were resuspended in 200 μl of 2%FCS/PBS to be acquired the following day. Data were acquired using a Cytek Aurora 5, and subsets identified as

shown in Supplementary Fig. 4A, B, 7A and 22A. To identify CD16 monocytes, an alternative gating strategy based on CCR2 and CD33 expression[168], due to the rapid down regulation of CD16 in cultured cells.

## Flow cytometric ex vivo cell phenotyping of DEGs

PBMCs were thawed in additional malaria study patients as above, at day 0, and day 28 post malaria infection. Due to sample limitations, not all patients had paired samples for this analysis. γδT-B-NK ex vivo cell phenotyping contained 11 day 0 samples and 17 day 28 (paired observations, $n = 8$) and CD4 ex vivo cell phenotyping contained 7 day 0 samples and 9 day 28 (paired observations, $n = 7$). Surface markers, dead cell stains and intracellular stains were performed at the concentrations provided with antibodies purchased from Becton-Dickson Biosciences, Biolegend or Invitrogen (γδT-B-NK: Supplementary Table 10 and CD4: Supplementary Table 11). CD4 ex vivo cell stains were rested for 2 hours prior to staining protocol. PBMCs were stained in Fc block with CD366 (TIM-3) and CD223 (LAG-3) at 37 °C for 90 minutes in γδT-B-NK ex vivo and with CD223 (LAG-3), CD49b and CCR7 at 37 °C for 45 minutes in CD4 ex vivo. Cells were then washed and stained at RT for 15 minutes with LIVE/DEAD Fixable Blue Dead Cell Stain, washed twice with 2% FCS/PBS and stained at RT for 30 minutes for additional surface markers. Following 2 washes with 2% FCS/PBS, cells were permeabilised with eBioscience Fixation/Permeabilization solution for 20 minutes on ice. Intracellular staining (ICS) was performed for 30 minutes on ice following washes to assess intracellular proteases and glycoproteins. Cells were fixed with BD stabilising fixative then resuspended in 200 µl of 2%FCS/PBS until acquisition. Data were acquired using a Cytek Aurora 5 and subsets and marker expression identified as described in Supplementary Figs. 2, 7A, B, 11B, C, 15A, and 20B.

## Flow cytometric data analysis

Flow cytometry data were analyzed in FlowJo version 10. Gating strategies are outlined in Supplementary Fig. 2, 4A, B, 7A, B, 11B, C, 15A, 20B and 22A. To measure the co-expression of cytokines or other markers on specific subsets, expression was analysed by SPICE (Simplified presentation of incredibly complex evaluations[49]), and permutation tests between combinations of cytokines/markers performed. For myeloid cytokine analysis, data was not included in analysis if <50 cell events were gated.

## Statistical analysis

All statistical analysis was performed in RStudio (R version 4.0 or greater). All statistical tests are two-sided. To assess correlations between cellular clusters identified by scRNAseq and flow cytometry, Pearson correlations were calculated. For cell proportions and expression levels, for paired data, Wilcoxon signed-rank test was used and for unpaired data, Mann-Whitney U test was performed. Graphing was done with ggplot2 v 3.4.1[169].

## Reporting summary

Further information on research design is available in the Nature Portfolio Reporting Summary linked to this article.

## Data availability

All data generated or analysed during this study are included in this article and supplementary information files or from the corresponding author upon request. The raw sequencing data and processed Cell Ranger outputs used in this study have been deposited in the NCBI data base under accession code GSE217930 https://www.ncbi.nlm.nih.gov/geo/query/acc.cgi?acc=GSE217930

The processed single cell RNA seq data are available in the Zenodo data base, here "annotated_Sabah_data_21Oct2022.rds"

https://doi.org/10.5281/zenodo.6973241. All other data generated in this study are provided in Supplementary Information. Source data are provided with this paper.

## Code availability

The code generated during this study is available at https://doi.org/10.5281/zenodo.8378626, https://github.com/MichelleBoyle/scRNAseq_malaria_2023

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

## Acknowledgements

We thank all the participants and parents of guardians involved in the clinical studies, along with the Malaysian Ministry of Health hospital directors and clinical staff at Kudat, Kota Marudu and Pitas district hospitals and at Queen Elizabeth Hospital, Kota Kinabalu. We thank support staff in QIMR Flow Cytometry and Imaging Facility, QIMR Sample Processing and Sequencing Service, and Dr. Jessica Engel for laboratory support. This work was supported by the National Health and Medical Research Council of Australia (Career Development Fellowship 1141632 to M.J.B., Ideas Grant 1181932 to M.J.B., Program Grants 1037304 and 1132975 to NMA, Senior Principal Research Fellowship 1135820 to NMA and by The Australian Centre of Research Excellence in Malaria Elimination Seed Grant to J.R.L.

## Author contributions

T.C., J.L., M.J.B. designed research study; N.D., J.L., K.B., D.A., A.S.N. conducted experiments; N.D., T.C., Z.P., J.L., J.H., D.A., M.S., M.J.B. analysed data; K.P., T.W., B.B., M.G., N.A. conducted and supervised the clinical studies and sampling; N.D., T.C., J.H., D.A., M.S., M.J.B. generated figures; N.D., T.C., Z.P., J.L., J.H., D.A., M.S., M.J.B. verified underlying data; A.L., C.E., M.J.B. provided supervision for staff and students and contributed to study design, interpretation and contextualization of data; N.D., M.J.B. led manuscript preparation with feedback from all other authors. All authors have read and approved the final version of the manuscript. M.J.B. approves this version of the manuscript on behalf of TC (deceased).

## Competing interests

The authors declare no competing interests.

## Additional information

**Peer review information** : *Nature Communications* thanks Christopher Sundling, Vanessa Amana-Bokagne, Thomas Jacobs and the other, anonymous, reviewer(s) for their contribution to the peer review of this work. A peer review file is available.

