## [Peer Review File · Nature Communications]

Malaria drives unique regulatory responses across multiple immune cell subsetsREVIEWER COMMENTS

Reviewer #1 (Remarks to the Author):

The study by Dooley et al. investigates the transcriptional immune landscape of PBMCs in the context of human malaria at single-cell resolution. The scRNA seq is based on 3 adults and 3 children with sample timepoints at acute disease, 7 days, and 28 days after treatment and 2 healthy controls from the same low-transmission area in Malaysia. The subsequent analysis for all immune cell subsets is based on the differential gene expression between the acute (day0) and day28 samples, used to comprehensively assess the effect of a natural infection on different immune cell subsets.

The study is very comprehensive and well-written with nice visualization of the data. The authors also connect their observations very well to the existing literature. This is, to my knowledge, the first time a study uses scRNA seq to broadly profile PBMCs at several timepoints after acute human malaria, hence providing a valuable resource for further investigation and follow-up studies.

Major points:

1. The authors describe the overall immune response to acute malaria in their cohort as a “tolerogenic responses”. Are uncomplicated malaria cases automatically due to a tolerogenic response?
2. The study uses individuals with “uncomplicated malaria”. Compared to severe malaria, “uncomplicated” comprises a broad spectrum of disease severity without a clear definition. It would be beneficial for the reader to provide more clinical information about the patients such as fever/body temperature, crp levels, blood cell counts, liver function, which anti-malaria drug.
3. Additional information about factors that might confound the analysis, such as CMV status (PMID: 25594173, especially regarding adaptive NK cells PMID: 21825173) and sickle cell trait genotype would be valuable.

4. The total number of cells in the data set (106 076 quality cells) to understand malaria driven immune responses stated in the manuscript might be misleading since only ~15% of these samples come from malaria disease samples. As most of the presented results are based on the comparison day0 (5 individuals, ~16 200) vs. day28 (6 individuals, ~34 000 cells; 8 individuals, ~55 000 cells if controls were included here), the authors should state this more clearly. It could be worth to down-sample the number of cells from day28 samples to the same number of cells at day0 to prove the robustness of the findings (see also point 11).

5. In this study, sample timepoint day28 is considered convalescence. When looking at the day28 vs endemic control comparison (Tab. S3E), there are still a large number of DEGs. In a recent study that described the immune landscape after natural malaria over one year it was shown that for example NK cell subsets are not back at normal levels after 1 month (PMID: 35443186). To support the identification of samples as convalescent, the authors could perhaps show (on a global level) that D28 samples are similar to the healthy controls.

6. Fig. 2, Fig. 3, Fig. 5, Fig. 6 gene expression heatmaps show $-\log(pval)$ but Fig.4 has \log_p_cat . What is the reason for this difference? Additionally, what is the reason for not correcting for multiple testing as is standard in transcriptomic analysis (e.g. by FDR) and hence not showing adjusted pvalues.

7. Regarding scRNA seq data presentation, the reader would benefit from being able to assess quality control plots in the supplementary material, indicating successful data integration of all samples/time points for all PBMCs, but also for the re-clustered data for each of the cell subsets, to exclude possible donor effects when clustering. It would also be good to add information about the number of cells used for comparing the DEGs for clusters/subclusters.

8. It would be helpful if the authors would provide their analysis code (for example via a GitHub repository) to make the analysis more transparent and easier to evaluate. This is especially important considering the aim of publishing the data as a resource.

9. I have not seen this type of ex-vivo assessment of myeloid cells used previously and found it relatively surprising that the cells produced so much cytokine without stimulation at day 28.

Is there a reason for not also performing stimulation with e.g. TLR-ligands or iRBC to assess function? The authors indicated such methods had been used before to assess tolerance in malaria.

10. When comparing cells between two time-points at single-cell granularity (such as Fig 6G), downsampling to the same number of cells would prove the robustness of the analysis. Further robustness could also be achieved by using the same cell number from each donor. This would prevent a potential bias of the statistical test due to a higher number of cells in one group/donor and the nature of single cells data (not all cells express the gene).

Minor points:

1. Line 42-43 please update numbers with latest WHO malaria report to report from 2022

2. Line 47-50 and 50-51 these statements would benefit from referencing to original research papers or reviews where this is further discussed.

3. Line 91-96 is not needed to introduce the study and feels a bit out of scope.

4. Line 100 “tolerogenic response during acute infection...” – it is somewhat unclear if this is solely a tolerogenic response since there is no clear comparison to non-tolerogenic/severe response.

5. Fig. 1E what proportion is calculated here?

6. The downregulation of cytokine and chemokine responses in the context of immune modulation/tolerogenic response but upregulation of receptors for antibody interaction (line 228-233) is further supported by recent findings (PMID: 35443186).

7. Fig. 5A requires axis labels (UMAP1, UMAP2)

8. Line 594/Legend Fig.6 – “P-value calculated using Kruskal wallis and post-Dunn test (FDR adjusted) indicated.” A bit unclear what is FDR adjusted.

9. Line 773-776 – please confirm the order of analysis steps. Up to my knowledge, PCs are used for dimensionality reduction of the data, followed by creating a nearest neighbor graph based on some distance in PCA space. This graph is then used for clustering (add which clustering algorithm used). For final visualization, non-linear dimensional reduction is

then run on (n) PCs and used to visualize the clusters found in the nearest neighbor graph. If this was the procedure, please update so it becomes more clear.

10. Did the clustering algorithm detect 15 clusters or more? If the 15 clusters are based on the merging of clusters, what was used for deciding the merging?

11. Please provide further descriptions for the sub-clustering procedure.

12. Which R package was used for gene set enrichment analysis?

13. For the DEG Day0 vs D28, were the 2 control samples also included?

14. Include a reference for offset plot package

15. Include a reference for ggplot package

16. The parasitemia values mentioned in the supplementary tables are indicated without any unit.

Reviewer #2 (Remarks to the Author):

Malaria is a parasitic infection causing significant burdens in the world especially, among children. Due to the complexity of the parasite, and its ability to evade the immune response, it is challenging for individuals to develop long-lasting immunity.

In this regard, Dooly et al investigated transcriptional changes occurring in immune subsets among children and adults diagnosed with acute uncomplicated *P. falciparum* malaria infection in a low-transmission setting in Malaysia.

A small number of patients with acute malaria were recruited and studied for several immune parameters during *falciparum* malaria and convalescence. Healthy control (individuals with no malaria) were also included. The analysis included ScRNA sequencing of isolated PBMCs cell population.

This study concludes that *P. falciparum* malaria induced tolerogenic immune cell response which protects the individual from inflammatory mediated immunopathogenesis, which in turn hampered the development of effective anti-parasitic immunity.

Many attempts to understand the immune mechanisms driven by the malaria parasites have been done before, making this study not the first of his kind. However, the manuscript presents some novel insights into the immunopathogenesis induced by *P. falciparum* malaria.

Though the manuscript is interesting, several points need to be clarified to ensure the validity of the results and improve its quality.

As general comments:

The authors did not consider patients' parasitemia throughout the experiment whereas it is known that some individuals may have delayed parasite clearance or relapses. The healthy control parasitemia was to be taken throughout to avoid any bias. The small sample size used and the disparity in the study participants make it difficult to generalize and validate the findings. The age and sex factors should have also been considered in reporting the results.

Additional patients were recruited for ex vivo cytokine production analysis and ex vivo cell phenotyping. How was the recruitment done? Have they also received treatment? More information about this cohort is needed.

In figure 1E, the authors reported a marked increase in CD4 T-cells and proliferative cells and a marked decrease in NK cells and $\gamma\delta$ T cells. What was the level of significance set for the p-values? Was the change in immune parameters person specific or analyzed in bulk? On the same figure, the author should explain in the legend what "unknown" stands for. The variation in the immune response among malaria groups and controls is not clear, we don't know if the increase or decrease in immune parameters is with respect to days or with respect to controls. Also, the malaria period and the convalescent period should be clearly defined.

Specific comments:

- What was the rationale for screening for immune regulation on day 7 and day 28?
- Which drugs were administered to the patients and for how long was the treatment?
- What were the inclusion and exclusion criteria of the study?
- How was the sample size determined?
- How did you ensure the healthy controls were not infected at any point in the study?
- How did you ensure a patient has passed from the stage of acute infection to convalescent?
- line 227, Figure 2G is mentioned in the text but is missing in the figure section
- lines 794 and 805, the tables S12 and S13 are not matching with the text

Reviewer #3 (Remarks to the Author):

Dooley et al. present a largely single-cell RNAseq-based analysis of PBMC from 6 patients diagnosed with malaria at initial hospital presentation, and then at 7 and 28 days after anti-malarial drug treatment. The authors present an analysis of DEGs in monocytes and dendritic cells, NK cells, gamma/delta T cells, CD4 T cells, and B cells. This analysis is supplemented with some flow cytometry-based analysis, in most cases to confirm select expression patterns. The authors report, broadly, the promotion of 'immunosuppressive' programs across these different cell types by malaria infection, and link this with the action of type I IFN signaling.

The strength of the manuscript is the breadth of analysis across cell types at the single cell level, and spanning at least 2 timepoints in a clinically relevant patient cohort. However, the weakness of the manuscript is rigor supporting the conclusions on the nature of the DEGs, which in this reviewer's opinion makes the study of limited impact to the field in terms of reflecting insight into the underlying mechanisms of disease/immunoregulation.

Furthermore, the significance of the study in terms of noteworthy results seems to lie chiefly in the identification of the Bregs, as the authors point out many examples in the literature (there are over 150 references) which are supported by the DEG patterns in the manuscript.

Major weaknesses

1) In the analysis of T cells, the expression of co-inhibitory receptors on T cells during active malaria infection versus their downregulation at day 28 is taken as evidence of a 'suppressive program'. However, several of these markers, including PD1, Lag3, etc. are also upregulated on activated effector cells. It is not clear from the analysis presented whether a 'suppressive' program unique to malaria infection is occurring versus a more typical activation program. Some of these same considerations apply to analysis of the other cell types. Without supportive data the analysis presented is thus highly speculative.

2) The analysis of both B and T cells suffers from not being done at a level of antigen specific lymphocytes responding to malaria. Especially when it comes to relatively rare subsets, such as the IL-10+ Bregs, it is thus impossible to discern whether or not malaria infection is

‘inducing’ these cells versus that the cells become more prominent because of other changes at the population level resulting in shifts in the pattern of heterogeneity within each subset. Or are the authors arguing for a state of generalized suppression within each subset, irrespective of whether or not cells are actively responding to infection?

3) Type I IFN signaling hallmarks are increasingly found to mark subsets of responding lymphocytes when assayed at the single cell level, and in many cases these DEGs correlate with increased or specialized function versus a suppressed state. Thus, the authors’ conclusion that a type I IFN response is responsible for the ‘suppressed’ signatures observed is again highly speculative.

4) The authors state in the conclusion that a weakness of the study is that it was not done at the level of the cells within an individual, which begs the question of the extent to which these general patterns hold across patients versus that they represent an artificial aggregate. Although the authors do confirm major, mostly phenotypic, findings at the protein level, these are mostly related to the idea of ‘suppressive’ programs, and again could be the result of comparing relatively activated populations to largely resting populations. The authors have also addressed this with their analysis of B cell IL-10 production, and the pattern indeed shows heterogeneity within the day 0 pool (which may be caused by several other variables given that the $n=8$ at both timepoints).

In the abstract, the authors state that their study a valuable ‘data set’ for further analysis. This reviewer agrees with that statement and recognizes the excellent technical achievements herein. However, to support the central interpretations of the DEG patterns that this study reveals, and their broader applicability to the field, additional studies are required.

Reviewer #4 (Remarks to the Author):

The present study by Dooley et al. provide a very interesting single cells sequencing data set, which across the different immune cell subsets analyze the impact of malaria. Although several aspects are already known, the present manuscript extends our view by showing the response of the various immune subsets and provide further evidence that protection from severe or clinical malaria has to be seen as a kind of tolerance induction against the

parasite. Thus it provides very important data which can be used to further understand the development of immunity in malaria. However I have some remaining questions:

Immunity or tolerance against malaria is dependent on the number of malaria infections. What was the rationale to choose samples from patients with a very different age, which might have had very different numbers of malaria episodes? And why the authors did not compare the immune response of children and adults? And along this line why the authors did not compare uncomplicated vs severe malaria?

The authors provide convincing evidence that the CD4 T cell response is dominated by Tr1 cells. However this signature disappears after treatment. Can the authors discuss if these Tr1 cells disappear or if it is more a disappearance of their signature which excludes a further identification after treatment?

In line 118 and Figure 1C and 1F the authors describe the annotation of CD8+ T cells and their DEGs. However this subset is not further analyzed at all. Since CD8+ T cells can also produce IL10 and might exert similar function as Tr1 cells it would be of interest to include also an analysis of them or even compare their profile with CD4+ Tr1 cells.

Specific points:

For cytokine analysis the secretion of cytokines was blocked and cells were then stained for cytokines but are they stimulated ex vivo antigen-specifically? Or do they still produce cytokines ex vivo since they saw antigen recently in acute malaria?

In line 227 the authors refer to a Figure 2G which is missing.

In line 670 the authors claimed that B cells are a major source of IL10. How they compared the amount of IL-10 produced by B cells to the amount of IL10 produced by other cell subsets?

I have difficulties to understand line 691 to 694. Why it is not possible to analyze data on an individual cell level?

We thank the reviews for positive and constructive feedback. We have addressed all the comments as outlined below and believe the manuscript has significantly improved. We hope that our manuscript will now be acceptable for publication.

We would like to highlight the following major changes in response to reviewer and editorial comment.

- 1) *Samples size* - we have added an additional 14 individuals to the data analysis (Fig2E/F, Fig3F/G/H, Fig4E/F/G, Fig5E/F/G, and Fig6H/I/J). In all cases this additional data supported and strengthened the previously reported findings.
- 2) *Definition of tolerogenic signature* - The tolerogenic signature is noted as such if identity DEGs had previously reported inhibitory functions (Attanasio 2016, ref 144 of manuscript, lines 947 methods). In revision, we have added new data of co-inhibitory receptor expression on CD4 T cell subsets at the protein level in additional study participants (Fig 5E, Fig S12). This data shows that the proportion of Tr1 and Th1 cells expressing co-inhibitory during malaria, and the levels of expression of these markers is equivalent or higher than FoxP3+ T-regulatory CD4 T cells. Expression of co-inhibitory markers on FoxP3+ T-regulatory cells is accepted to be indicative of suppressive function. Together data is supportive of the hypothesis that upregulation of co-inhibitory receptors on Tr1 and Th1 cells during malaria infection is not due to transient activation but is instead indicative of tolerogenic potential of these cells.
- 3) *Clinical reporting* - we have added additional clinical parameters that were available including drug treatments received, self-reported previous malaria infection, fever temperature, white blood cell count, hematocrit. In addition, all patients have been tested for cytomegalovirus (CMV) infection, with 100% being CMV positive. These parameters are now included in Supplementary Table S1. Additionally, we have extensively edited the methods to clarify the source and selection of clinical samples used in this study (lines 867-903).
- 4) *Clarification of data analysis approach* - The main aim of our study was to understand the transcriptional activation programs induced in malaria on specific cell subsets. To achieve this, PBMCs were first analysed into 15 high level cell clusters, and then each cell subset further analysed to identify specific subsets. This resulted in relatively low cell numbers in some cell subsets, and as such were not able to further analyse transcriptional changes at the individual donor level. In revision, clarification of this approach has been made throughout the manuscript. In addition, in revision we have presented in Supplementary Tables S4, S7, S10, S13, S16 and S19 the cell numbers in each subsets from each individual (visually represented in Figure S23), and have included Supplementary Figures (S3, S5, S12, S14, S19 and S21) with the expression of DEGs at day 0 and day 28 at the individual donor level. This approach is comparable to previously published analysis, for example of subset specific changes during covid and influenza Zhu et al, Immunity 2020.
- 5) *Analysis of CD8 T cells* - Further analysis of CD8 T cell responses is now included and outlined in lines 557-576 of the manuscript and in Figure S19. DEGs in CD8 T cells only

had limited overlap with the DEGs expressed in Tr1 CD4 T cells, and we did not detect upregulation of IL10 in CD8 cells.

All additional comments by reviewers have been addressed as outlined below.

REVIEWER COMMENTS

Reviewer #1 (Remarks to the Author):

The study by Dooley et al. investigates the transcriptional immune landscape of PBMCs in the context of human malaria at single-cell resolution. The scRNA seq is based on 3 adults and 3 children with sample timepoints at acute disease, 7 days, and 28 days after treatment and 2 healthy controls from the same low-transmission area in Malaysia. The subsequent analysis for all immune cell subsets is based on the differential gene expression between the acute (day0) and day28 samples, used to comprehensively assess the effect of a natural infection on different immune cell subsets.

The study is very comprehensive and well-written with nice visualization of the data. The authors also connect their observations very well to the existing literature. This is, to my knowledge, the first time a study uses scRNA seq to broadly profile PBMCs at several time-points after acute human malaria, hence providing a valuable resource for further investigation and follow-up studies.

Response: We thank the review for the helpful feedback of our manuscript. We have addressed all queries as outlined below.

Major points:

1. The authors describe the overall immune response to acute malaria in their cohort as a 'tolerogenic responses'. Are uncomplicated malaria cases automatically due to a tolerogenic response?

Response: Uncomplicated malaria can occur for a number of reasons, including tolerized responses to the parasite, but also other host factors such as an adaptive antibody response that can control parasite growth, genetic susceptibility, host age during infection. Parasite virulence of infecting strain can also contribute to disease severity. We have added a clarification of this point in lines 43-46 of the introduction.

2. The study uses individuals with uncomplicated malaria. Compared to severe malaria, 'uncomplicated' comprises a broad spectrum of disease severity without a clear definition. It would be beneficial for the reader to provide more clinical information about the patients such as fever/body temperature, crp levels, blood cell counts, liver function, which anti-malaria drug.

Response: We agree with the reviewer that uncomplicated malaria includes a broad spectrum of disease severities and phenotypes. In revision we have added the following clinical details; drug treatments received, self-reported previous malaria infection, fever temperature, white cell count, hematocrit. Liver function and CRP levels are not available. See Supplementary Tables S1. We have also noted as a limitation that we were unable to assess the impact of disease manifestation on transcriptional data in discussion line 830-832.

3. *Additional information about factors that might confound the analysis, such as CMV status (PMID: 25594173, especially regarding adaptive NK cells PMID: 21825173) and sickle cell trait genotype would be valuable.*

Response: We agree that CMV and sickle cell trait genotypes may be important confounders of our findings. In revision, we have now tested the CMV status of individuals, which is included along with other clinical data in Supplementary Tables S1. All individuals were sero-positive for CMV. Genetic traits, including sickle cell are not available within these cohorts, and this is now included as limitation in discussion, lines 838-839.

4. *The total number of cells in the data set (106 076 quality cells) to understand malaria driven immune responses stated in the manuscript might be misleading since only ~15% of these samples come from malaria disease samples. As most of the presented results are based on the comparison day0 (5 individuals, ~16 200) vs. day28 (6 individuals, ~34 000 cells; 8 individuals, ~55 000 cells if controls were included here), the authors should state this more clearly. It could be worth to down-sample the number of cells from day28 samples to the same number of cells at day0 to prove the robustness of the findings (see also point 11).*

Response: We agree that the total cell number of analysed cells could be misleading. We have now provided clarity on the numbers of cells used in the day 0 v day 28 analysis in lines 142 of results and modified the discussion lines 717. We also agree with the reviewer that to increase robustness of the analysis, it would be ideal to down sample to the same cell number, or to analyze each individual separately (as subsequently discussed in point 10). However, due to the relatively low cell numbers in each cell subset once annotated and in subsets, we have found that this down sampling greatly reduces the ability to detect changes. To address this issue, in revision we have presented in Supplementary Tables S4, S7, S10, S13, S16 and S19 the cell numbers in each subsets from each individual, and have included Supplementary Figures (S3, S5, S12, S14, S19 and S21) with the expression of DEGs at day 0 and day 28 at the individual donor level. This data shows that the DEG expressions are largely consistent across donor.

5. *In this study, sample timepoint day28 is considered convalescence. When looking at the day28 vs endemic control comparison (Tab. S3E), there are still a large number of DEGs. In a recent*

study that described the immune landscape after natural malaria over one year it was shown that for example NK cell subsets are not back at normal levels after 1 month (PMID: 35443186). To support the identification of samples as convalescent, the authors could perhaps show (on a global level) that D28 samples are similar to the healthy controls.

Response: The global level differences between day 28 and endemic ‘healthy’ control cells are now presented in Sup Figure S1E. While there were a significant number of DEGs between day 28 and endemic controls, this may be either due to sustained changes due to malaria infection, or alternatively due to individual baseline heterogeneity. These possibilities are now noted in results lines 135-138 and as a limitation of the study in discussion lines 825-828. As we cannot exclude the differences being due to individual heterogeneity, we chose to focus on the day0/28 comparison.

6. Fig. 2, Fig. 3, Fig. 5, Fig. 6 gene expression heatmaps show -log(pval) but Fig.4 has log_p_cat. What is the reason for this difference? Additionally, what is the reason for not correcting for multiple testing as is standard in transcriptomic analysis (e.g. by FDR) and hence not showing adjusted pvalues.

Response: We thank the reviewer for picking up this error. In all figures the p values are $-\log(\text{adjPval})$, with p values adjusted with Benjamini-Hochberg FDR for multiple comparisons. These errors have been corrected in figures and adjustment method noted in methods line 965-967.

7. Regarding scRNA seq data presentation, the reader would benefit from being able to assess quality control plots in the supplementary material, indicating successful data integration of all samples/time points for all PBMCs, but also for the re-clustered data for each of the cell subsets, to exclude possible donor effects when clustering. It would also be good to add information about the number of cells used for comparing the DEGs for clusters/subclusters.

Response: We have now included a supplementary figure of all pre-processing steps of data in Sup Figure S1. This includes (A) Pre- and Post-filter violin plots of nUMI (transcript counts per cell), nGene (gene counts per cell), $\log_{10}\text{GenesPerUMI}$, mitoRatio (percentage of mitochondrial genes per cell). Cells removed based on filters $\text{nUMI} > 500$, $\text{nGene} > 250$, $\log_{10}\text{GenesPerUMI} > 0.8$, $\text{mitoRatio} < 0.2$. (B) Barplots of mitoRatio and numbers french each sample to justify removal of sample “child1day0” due to low mitoRatio and low no. of cells. (C) UMAPs of sample distribution pre- and post-integration by donor. Integrated UMAP of cluster distribution and numbering based on Seurat default shared nearest neighbor (SNN) clustering algorithm at a resolution of 0.6. Annotated cluster UMAP to define cell subset distribution in single cell transcriptional data. (D) Identification and group of annotated cell lineages based on expression of canonical genes in dotplot. Further, supplementary Figure S23 includes the UMAP visualisation of PMBCs and all re-clustered data at the individual donor level.

Additional, for PBMC clusters and sub-clustered cell subsets, we report the number of cells in each clusters/subclusters, including the breakdown of these cells at the individual level, in Supplementary Tables S4, S7, S10, S13, S16 and S19. We agree that this information provides important context to the data.

8. *It would be helpful if the authors would provide their analysis code (for example via a GitHub repository) to make the analysis more transparent and easier to evaluate. This is especially important considering the aim of publishing the data as a resource.*

Response: Analysis code is now available
https://github.com/MichelleBoyle/scRNAseq_malaria_2023.
This link is listed in manuscript lines 1060.

9. *I have not seen this type of ex-vivo assessment of myeloid cells used previously and found it relatively surprising that the cells produced so much cytokine without stimulation at day 28. Is there a reason for not also performing stimulation with e.g. TLR-ligands or iRBC to assess function? The authors indicated such methods had been used before to assess tolerance in malaria.*

Response: In our experience, when using PBMC samples from individuals with a current or recent infection, cytokine production from myeloid cells is far higher than that seen in healthy controls taken from Australian donors. This relatively high cytokine production may be due to recent malaria, or alternatively due to other environmental or host factors between endemic and non-endemic populations. Understanding the mechanisms of these differences is the focus of other projects in our lab. In the current study, we have assessed cytokine production in myeloid cells *ex vivo*, rather than following stimulation with TLR-ligands or parasites, as we aimed to analyze cells at infected and convalescence timepoints in the same manner as the transcriptional data, which was taken from unstimulated PBMCs.

10. *When comparing cells between two time-points at single-cell granularity (such as Fig 6G), downsampling to the same number of cells would prove the robustness of the analysis. Further robustness could also be achieved by using the same cell number from each donor. This would prevent a potential bias of the statistical test due to a higher number of cells in one group/donor and the nature of single cells data (not all cells express the gene).*

Response: The primary aim of this study was to dissect malaria associated transcriptional changes at the cell subset level. While a large number of PBMCs were analyzed, once sub-clustering is performed, some cell subsets have relatively low cell numbers to use in DEG identification between day 0 and day 28. Attempts to down sample to include same cell numbers per individual significantly reduced power to detect malaria associated changes. To address this limitation, we have now included details of cell numbers/donor/cluster in Supplementary Tables S4, S7, S10, S13, S16 and S19. Additionally, expression of all DEGs that were identified with all donor data at day0 and 28, are now presented at the individual level for all cell subclusters in Supplementary Figures S3, S5, S12, S14, S19

and S21. This approach is comparable to previously published analysis, for example of subset specific changes during covid and influenza Zhu et al, Immunity 2020. The inability to analyse data at the individual donor level is noted in limitations, discussion lines 821-825.

Minor points:

1. Line 42-43 please update numbers with latest WHO malaria report to report from 2022

Response: Updated, lines 43, reference #1

2. Line 47-50 and 50-51 these statements would benefit from referencing to original research papers or reviews where this is further discussed.

Response: Citations #4 Doolan et al, Clin Microbiol Rev 2009 and #5 Beeson et al, Sci Transl Med, 2019 have been added to these sections.

3. Line 91-96 is not needed to introduce the study and feels a bit out of scope.

Response: These lines have been removed.

4. Line 100 *tolerogenic response during acute infection* it is somewhat unclear if this is solely a tolerogenic response since there is no clear comparison to non-tolerogenic/severe response.

Response: Lines referring to a tolerogenic response have been edited for clarification.

5. Fig. 1E what proportion is calculated here?

Response: We have added the clarification “as a proportion of total analysed cells within each individual” to line 126 and to the figure legend.

6. The downregulation of cytokine and chemokine responses in the context of immune modulation/tolerogenic response but upregulation of receptors for antibody interaction (line 228-233) is further supported by recent findings (PMID: 35443186).

Response: We have added this reference to this section (lines 211-213), and to discussion lines 738-741.

7. Fig. 5A requires axis labels (UMAP1, UMAP2)

Response: This figure has been corrected

8. Line 594/Legend Fig.6 *P-value calculated using Kruskal wallis and post-Dunn test (FDR adjusted) indicated.* A bit unclear what is FDR adjusted.

Response: The FDR adjusted p values refers to the post-Dunn test across multiple cell groups. This clarification is added to the figure legend.

9. Line 773-776 ??? please confirm the order of analysis steps. Up to my knowledge, PCs are used for dimensionality reduction of the data, followed by creating a nearest neighbor graph based on some distance in PCA space. This graph is then used for clustering (add which clustering algorithm used). For final visualization, non-linear dimensional reduction is then run on (n) PCs and used to visualize the clusters found in the nearest neighbor graph. If this was the procedure, please update so it becomes more clear.

Response: The methods section has been edited for clarity (lines 941-949).

10. Did the clustering algorithm detect 15 clusters or more? If the 15 clusters are based on the merging of clusters, what was used for deciding the merging?

Response: Clustering identified 23 clusters, which were merged to the 15 clusters in a supervised manner. The original and merged clusters are now shown in additional Supplementary Figure S1.

11. Please provide further descriptions for the sub-clustering procedure.

Response: The sub-clustering procedure was further detailed in the methods section in lines 949-954.

12. Which R package was used for gene set enrichment analysis?

Response: GSEA was completed using analysis software available <https://www.gsea-msigdb.org/gsea/index.jsp>. This information is now included in lines 967-973.

13. For the DEG Day0 vs D28, were the 2 control samples also included?

Response: For all DEGs identified between day 0 and day28, control cells were not included. This is now clarified in results lines 140-142.

14. Include a reference for offset plot package

Response: R package and reference has been added to methods line 973-974.

15. Include a reference for ggplot package

Response: R package and references has been added to methods line 1030.

16. The parasitemia values mentioned in the supplementary tables are indicated without any unit.

Response: Units are now indicated in supplementary table S1.

Reviewer #2 (Remarks to the Author):

Malaria is a parasitic infection causing significant burdens in the world especially, among children. Due to the complexity of the parasite, and its ability to evade the immune response, it is challenging for individuals to develop long-lasting immunity.

*In this regard, Dooly et al investigated transcriptional changes occurring in immune subsets among children and adults diagnosed with acute uncomplicated *P. falciparum* malaria infection in a low-transmission setting in Malaysia.*

A small number of patients with acute malaria were recruited and studied for several immune parameters during falciparum malaria and convalescence. Healthy control (individuals with no malaria) were also included. The analysis included ScRNA sequencing of isolated PBMCs cell population.

*This study concludes that *P. falciparum* malaria induced tolerogenic immune cell response which protects the individual from inflammatory mediated immunopathogenesis, which in turn hampered the development of effective anti-parasitic immunity.*

*Many attempts to understand the immune mechanisms driven by the malaria parasites have been done before, making this study not the first of his kind. However, the manuscript presents some novel insights into the immunopathogenesis induced by *P. falciparum* malaria.*

Though the manuscript is interesting, several points need to be clarified to ensure the validity of the results and improve its quality.

Response: We thank the reviewer for the helpful review of our manuscript. We have addressed all comments as below and addressed concerns have strengthened the manuscript.

As general comments:

The authors did not consider patients' parasitemia throughout the experiment whereas it is known that some individuals may have delayed parasite clearance or relapses. The healthy control parasitemia was to be taken throughout to avoid any bias. The small sample size used and the disparity in the study participants make it difficult to generalize and validate the findings. The age and sex factors should have also been considered in reporting the results.

Response: We agree with the reviewer that considering confounding factors such as parasitemia clearance, age and sex are important in interpreting our data. In the revised manuscript we have provided additional clinical information for all individuals including drug treatments received, self reported previous malaria infection, fever temperature, white cell count, and hematocrit (see Supplementary Table S1). Of note, while clinical characteristics varied, none of the individuals used for scRNAseq analysis reported a previous malaria episode.

Regarding the concern that some individuals may have delayed parasite clearance, within our study all participants successfully cleared parasite infection following. This information, along with expanded details on the clinical cohorts is included in revised manuscript methods lines 867-903.

To address concerns of small sample sizes, we have added an additional 14 study participants to data presented in Figures 2E-G, 3F-H, 4E-F, 6E-J. We have analysed this

data in relation to age, sex and parasitemia and included these results in Supplementary Figures S4, S8-10, S17, S17 and S22.

Additional patients were recruited for ex vivo cytokine production analysis and ex vivo cell phenotyping. How was the recruitment done? Have they also received treatment? More information about this cohort is needed.

Response: We apologize for the lack of clarity regarding these study cohorts in the original submission. In revision, we have significantly expanded the description of the study cohorts which provided PBMCs for this immunology analysis. All individuals for both scRNAseq analysis, and subsequent *ex vivo* cytokine/phenotyping analysis were collected within the same parent studies and clinical sites. Details of recruitment and treatment are now described in lines 867-903, and available clinical data is presented in Supplementary Table S1.

In figure 1E, the authors reported a marked increase in CD4 T-cells and proliferative cells and a marked decrease in NK cells and ??? T cells. What was the level of significance set for the p-values?

Response: The text regarding changes to cell proportions during infection has been updated to reflect that not all changes reached $p < 0.05$ (see lines 124-126). All comparisons and p values are now included in text for clarity.

Was the change in immune parameters person specific or analyzed in bulk?

Response: Throughout the manuscript, DEGs for cell types and subsets are calculated on total cells/day (not person specific). We were not able to analyse DEGs at the individual level due to the low number of contributing cells in some individuals/clusters. Cell number information for each individual/cluster is now included in Supplementary Tables S4, S7, S10, S13, S16 and S19. Additionally, expression of all DEGs that were identified with all donor data at day 0 and 28, are now presented at the individual level for all cell subclusters in Supplementary Figures S3, S5, S12, S14, S19 and S21. This approach is comparable to previously published analysis, for example of subset specific changes during covid and influenza Zhu et al, Immunity 2020. The inability to analyse data at the individual donor level is noted in limitations, discussion lines 821-825.

On the same figure, the author should explain in the legend what ???unknown??? stands for.

Response: The unknown cluster are cells that did not express any known cell lineage markers and relatively high expression of mitochondrial genes, possibly indicating low quality cells. In revision, we re-named this cluster 'uncharacterised', and added clarification of these cells to the text lines 119-120.

The variation in the immune response among malaria groups and controls is not clear, we don't know if the increase or decrease in immune parameters is with respect to days or with respect to controls. Also, the malaria period and the convalescent period should be clearly defined.

Response: We believe this comment is addressing confusion in the log fold change of differentially expressed genes (DEGs). As outlined in figure legends, these are differentially expressed genes between Day 0 and Day 28, in which fold changes greater than 0 (red) indicate genes upregulated at Day 0 and fold changes less than 0 (blue) indicate genes upregulated at Day 28.

Specific comments:

- What was the rationale for screening for immune regulation on day 7 and day 28?

Response: Samples for this study were from parent studies which had stored blood samples collected at day 0, 7 and 28 days post treatment. All individuals had successfully cleared parasite infection by 72 hours (3 days) post treatment. Future studies with more closely sampled timepoints will be required to understand the kinetics of immune cell activation at finer detail. The limited time points post treatment is now noted as a study limitation in lines 828-830.

- Which drugs were administered to the patients and for how long was the treatment?

Response: Details of drug treatments used in parent studies are now included in methods lines 877-882, and are listed for each individual used in the current study in Supplementary Table S1.

- What were the inclusion and exclusion criteria of the study?

Response: Inclusion/exclusion criteria for the parent studies is now described in lines 875-877. For the current study, samples were selected based on confirmed *P. falciparum* infection by PCR, and sample availability.

- How was the sample size determined?

Response: While sample size calculations were performed for the parent clinical studies, no sample size calculations were performed for the current immunology analysis. For scRNAseq analysis, the selection of 6 infected patients at 3 time points, and 2 uninfected healthy donor controls is consistent with single-cell transcriptional analysis in the field (For example see Szabo et al, Nat Comms 2019 | <https://doi.org/10.1038/s41467-019-12464-3> used 2 diseased and 2 healthy donors. Guo et al, Nat Comms 2020 | <https://doi.org/10.1038/s41467-020-17834-w> used 2 diseased individuals at 3 time points). For protein level analysis, sample sizes are consistent with ours and others similar studies of malaria immune responses which have used 8-15 individuals at repeated time points.

- How did you ensure the healthy controls were not infected at any point in the study?

Response: In the current study, healthy controls were sampled at a single time point. These individuals had been in the area for the preceding 3 weeks, were negative for *Plasmodium* by microscopy, and had no history of fever in the previous 48 hours. This information is included in lines 888-890.

- *How did you ensure a patient has passed from the stage of acute infection to convalescent?*

Response: All participants in the parent studies successfully cleared parasite infection within 72 hours of drug treatment. Parasite clearance was confirmed by thick smear or PCR. This information is now included in lines 878-879.

- *line 227, Figure 2G is mentioned in the text but is missing in the figure section*

- *lines 794 and 805, the tables S12 and S13 are not matching with the text*

Response: We apologize for these errors which have both been corrected in revision.

Reviewer #3 (Remarks to the Author):

Dooley et al. present a largely single-cell RNAseq-based analysis of PBMC from 6 patients diagnosed with malaria at initial hospital presentation, and then at 7 and 28 days after anti-malarial drug treatment. The authors present an analysis of DEGs in monocytes and dendritic cells, NK cells, gamma/delta T cells, CD4 T cells, and B cells. This analysis is supplemented with some flow cytometry-based analysis, in most cases to confirm select expression patterns. The authors report, broadly, the promotion of immunosuppressive programs across these different cell types by malaria infection, and link this with the action of type I IFN signaling.

The strength of the manuscript is the breadth of analysis across cell types at the single cell level, and spanning at least 2 timepoints in a clinically relevant patient cohort. However, the weakness of the manuscript is rigor supporting the conclusions on the nature of the DEGs, which in this reviewer's opinion makes the study of limited impact to the field in terms of reflecting insight into the underlying mechanisms of disease/immunoregulation. Furthermore, the significance of the study in terms of noteworthy results seems to lie chiefly in the identification of the Bregs, as the authors point out many examples in the literature (there are over 150 references) which are supported by the DEG patterns in the manuscript.

Response: We thank the reviewer for the helpful review of our manuscript. We have addressed concerns as outlined below, and with revisions from all other reviewers, feel that revision has significantly strengthened presentation of the data.

Major weaknesses

1) In the analysis of T cells, the expression of co-inhibitory receptors on T cells during active malaria infection versus their downregulation at day 28 is taken as evidence of a 'suppressive program'. However, several of these markers, including PD1, Lag3, etc. are also upregulated on activated effector cells. It is not clear from the analysis presented whether a 'suppressive' program unique to malaria infection is occurring versus a more typical activation program. Some of these same considerations apply to analysis of the other cell types. Without supportive data the analysis presented is thus highly speculative.

Response: Within Tr1 and Th1 CD4 T cells, we identify transcriptional upregulation of not only LAG3 and PD1 during malaria, but also multiple other co-inhibitory receptors (OX40, TNFR2, GITR, TIM3, CTLA4) and other genes with known functions in driving Tr1 or IL10 development (MAF, BLIMP1, STING1). The role of these multiple co-inhibitory receptors has been shown previously, and we have noted that we identify DEGs as tolerogenic based on existing data (Attanasio 2016, ref 145 of manuscript, lines 974-975 methods).

To support the finding of a suppressive CD4 T cell program in malaria, in revision we have added data of the co-inhibitory receptor upregulation on CD4 T cells at the protein level on additional study participants. This data identifies large proportions of Tr1 CD4 T cells during malaria infection (LAG3+CD49b+), which up regulate co-inhibitory markers CD120b, CTLA4, TIM3 and PD1. The level of expression of these regulatory proteins on Tr1 and Th1 cells was as high, or higher than that seen on FoxP3+ Treg cells, which are known suppressive CD4 T cells. As such, data is supportive of malaria inducing CD4 T cell responses with suppressive functions.

For other cell subsets, while PD1 and Lag3 are also associated with activated effector cells, our data shows that it is not only these two markers but multiple co-inhibitory receptors that are upregulated (for example, for NK cells we also detected OX40, CD137/41BB, GITR, Tim3, and for $\gamma\delta$ T cells, OX40 and Tim3). Review of the literature shows that this upregulation of multiple co-inhibitory receptors during infection have not been reported in comparable scRNAseq data sets of SARS-Co-V2, influenza, HIV or dengue (see paper references 38-40 and 117). We have noted the distinction of our data set from these previously published studies in discussion lines 798-801.

2) The analysis of both B and T cells suffers from not being done at a level of antigen specific lymphocytes responding to malaria. Especially when it comes to relatively rare subsets, such as the IL-10+ Bregs, it is thus impossible to discern whether or not malaria infection is 'inducing' these cells versus that the cells become more prominent because of other changes at the population level resulting in shifts in the pattern of heterogeneity within each subset. Or are the authors arguing for a state of generalized suppression within each subset, irrespective of whether or not cells are actively responding to infection?

Response: For CD4 T cells the proportional distribution of subsets identified by scRNAseq, including the regulatory Tr1 subset, did not change between acute infection (day 0) and convalescence (day 28) (Supplementary Fig S13). Instead, within Tr1 and Th1

cells, malaria was associated with a significant increase of a large number regulatory DEGs (including co-inhibitory receptors *LAG3*, *TNFRSF1B*, *CTLA4*, *TNFRSF4*, *TNFRSF18*, *HAVCR2*, *PRDM1*, and genes associated with Tr1 cell development *IL10*, *STING1*, and *MAF*). These DEGs were not seen in other CD4 T cell subsets. In revision, these transcriptional data were confirmed at the protein level on additional individuals (lines 528-556, Figure 5E-G, and Fig S15-17). As such, data is consistent with malaria inducing a regulatory program specifically within Th1/Tr1 cell subsets, and not due to a population level shift in CD4 T cell subset distribution. Identifying the antigen specificity of these regulatory responses is outside the scope of this manuscript, but will be investigated in future studies, and this limitation is now noted in lines 843-845.

For B cells, while there was an increased proportion of Plasmablasts identified within scRNAseq data during infection, the proportion of other B cell subsets was largely consistent between day 0 and day 28 (Supplementary Fig S20A). Additionally, HIF1A, which is a driver of IL10 production in B cells, was upregulated on multiple B cell subsets. As such, we measured IL10 production from the total B cell population. In revision, we have analyzed IL10 production at day 0 and day 28 in an additional 6 individuals. Median IL10 expression from B cells during malaria was ~5%. This data is analysed as a proportion of B cells, and as such is unlikely to be due to changes to other cell populations. Future studies are required to investigate whether these Bregs are responding in an antigen specific or globally suppressive manner, and this limitation is now noted in lines 843-845.

3) Type I IFN signaling hallmarks are increasingly found to mark subsets of responding lymphocytes when assayed at the single cell level, and in many cases these DEGs correlate with increased or specialized function versus a suppressed state. Thus, the authors' conclusion that a type I IFN response is responsible for the suppressed signatures observed is again highly speculative.

Response: We agree with the reviewer that Type I IFN signaling may not be responsible for the suppressive signature, and we have edited the manuscript throughout to make clear that this link is only one possibility and requires further studies to confirm. See edits in lines 813-817.

4) The authors state in the conclusion that a weakness of the study is that it was not done at the level of the cells within an individual, which begs the question of the extent to which these general patterns hold across patients versus that they represent an artificial aggregate. Although the authors do confirm major, mostly phenotypic, findings at the protein level, these are mostly related to the idea of suppressive programs, and again could be the result of comparing relatively activated populations to largely resting populations. The authors have also addressed this with their analysis of B cell IL-10 production, and the pattern indeed shows heterogeneity within the day 0 pool (which may be caused by several other variables given that the n=8 at both timepoints).

Response: The primary aim of this study was to dissect malaria associated transcriptional changes at the cell subset level. While a large number of PBMCs were analysed, once sub-

clustering is performed, some cell subsets have relatively low cell numbers to use in DEG identification between day 0 and day 28. To address the concern of artificial aggregate driving our data, we have now included details of cell numbers/donor/cluster in Supplementary Tables S4, S7, S10, S13, S16 and S19, and expression of all DEGs presented at the individual level for all cell subclusters in Supplementary Figures S3, S5, S12, S14, S19 and S21. This approach is comparable to previously published analysis, for example of subset specific changes during covid and influenza Zhu et al, Immunity 2020. The inability to analyse data at the individual donor level is noted in limitations, discussion lines 821-825.

As noted above, the suppressive programs here induced in malaria have not been identified previously in other diseases in comparable data sets. While we agree that the levels of suppressive programs, or IL10 expression for B cells may be influenced by other variables not measured here, the heterogeneity in these data is as expected from human studies. To strengthen our conclusions, we have increased the number of individuals within all phenotypic data sets and analysed all phenotypic level data to explore associations with age, parasitemia and sex, and these data are included in Supplementary Figures S4, S8-10, S16, S17 and S22.

In the abstract, the authors state that their study a valuable data set for further analysis. This reviewer agrees with that statement and recognizes the excellent technical achievements herein. However, to support the central interpretations of the DEG patterns that this study reveals, and their broader applicability to the field, additional studies are required.

Response: In revision, we have made major improvements to the manuscript including analysis of an additional 14 individuals, analysis of co-inhibitory markers on CD4 T cells at the protein level, and analysis of CD8 T cells. These data strengthen the conclusions of our study. In addition, careful review of the literature shows that the patterns of DEGs reported here in malaria have not been previously identified in comparable data sets of SARS-Co-V2, influenza, HIV or dengue. In providing this data set to the community, with accessible data, Seurat files and code, subsequent analysis and additional studies by others have the opportunity to support, or refute, our interpretations of the DEGs identified.

Reviewer #4 (Remarks to the Author):

The present study by Dooley et al. provide a very interesting single cells sequencing data set, which across the different immune cell subsets analyze the impact of malaria. Although several aspects are already known, the present manuscript extends our view by showing the response of the various immune subsets and provide further evidence that protection from severe or clinical malaria has to be seen as a kind of tolerance induction against the parasite. Thus it provides very important data which can be used to further understand the development of immunity in malaria. However I have some remaining questions:

Response: We thank the reviewer for helpful feedback on our manuscript. We have addressed comments and suggestions as outlined and believe that the manuscript is strengthened in review.

Immunity or tolerance against malaria is dependent on the number of malaria infections. What was the rationale to choose samples from patients with a very different age, which might have had very different numbers of malaria episodes?

Response: The aim of this study was to identify cell specific transcriptional profiles induced by malaria infection. Previous studies have shown that both anti-parasitic and anti-disease immunity are associated with both exposure and age independently (pmid: 30044224). We selected individuals across different ages to identify transcriptional profiles induced broadly by malaria independently of host age. This information is now included in methods lines 869-871. While in high transmission areas age and cumulative exposure are strongly associated, Sabah Malaysia is a low transmission area (incidence 0.18/1000 people in 2011), and all individuals used in scRNAseq analysis reported no prior malaria infections. This information is now included in Supplementary Tables S1. As such, data is likely consistent with similarly low prior exposure across ages.

And why the authors did not compare the immune response of children and adults ? And along this line why the authors did not compare uncomplicated vs severe malaria?

Response: We have not included an analysis of the children v adults' responses at acute infection as we are not confident in the robustness of that analysis due to low samples size comparing $n=2$ children, $n=3$ adults. All participants in the current study had uncomplicated malaria, and additional studies will be required to compare to severe disease. This limitation is now noted in lines 830-832.

The authors provide convincing evidence that the CD4 T cell response is dominated by Tr1 cells. However this signature disappear after treatment. Can the authors discuss if these Tr1 cells disappear or if it is more a disappearance of their signature which exclude a further identification after treatment?

Response: The frequency Tr1 cells identified transcriptionally by sub-clustering CD4 T cells (Figure 5A/B), does not change between infection and convalescent time points (as shown in Supplementary Figure S9A). However, during infection the regulatory signature of Tr1 cells is further increased (Fig 5D). These details have been clarified in lines 492-495. In revision, we have added additional analysis of CD4 T cells at the protein level in additional participants. When identified by protein level expression of LAG3 and CD49b (as we have previously, Edwards et al, JCI, 2023), Tr1 cells are increased at day 0 compared to day 28 in frequency, expression of T cell activation markers (CD38 and ICOS) and expression of regulatory surface proteins (CD120b, CTLA-4, TIM-3 and PD1) (Fig 5E-G). Together, this data suggests that while the underlying transcriptional program of Tr1 cells

is maintained, expression of inhibitory receptors such as LAG3 largely disappears following treatment.

In line 118 and Figure 1C and 1F the author describe the annotation of CD8+ T cells and their DEGs. However this subset is not further analyzed at all. Since CD8+ T cells can also produce IL10 and might exert similar function as Tr1 cells it would be of interest to include also an analysis of them or even compare their profile with Cd4+ Tr1 cells.

Response: Further analysis of CD8 T cell responses is now included. Compared to CD4 T cells, only low numbers of DEGs were identified and no regulatory cluster, or IL10 signature was detected. We have included the following in the manuscript in lines 557-575 and presented CD8 T cell responses in Sup Figure S18

“The role of CD8 T cells in immunity to *P. falciparum* blood stage malaria is unclear, particularly due to the lack of MHCI on the surface of RBCs. However, limited studies have reported activation and increased function of CD8 T cells during malaria, particularly in hospitalised patients (117). To investigate CD8 cell activation in acute malaria, CD8 T cells identified in PBMCs were subclustered. Naive/Central memory, Memory progenitor, Effector memory and Cytotoxic effector cells were identified (**Fig. S18A-B, Table S14**), The proportion of which was largely consistent between infected (day 0) and convalescence (day 28) (**Fig. S18C**). We conducted DEG analysis for each CD4 T cell subset, on day 0 (malaria) compared to day 28 (post treatment (**Table S15**, individual contribution to each subset was consistent (**Table S16**). Compared to CD4 T cells only a small number of DEGs between day 0 and 28 were identified, the largest number being in cytotoxic effector CD8 cells (**Fig. S18D, Table S15**). Upregulated genes included those relating to increased inflammation and cytotoxicity (*IFNG, TNF, CCL4, CCL3, IFNGR1, GZMB, PRF1*) (**Fig. S18E**). Genes associated with regulation were also upregulated in Effector subsets but no other CD8 subsets (including *LAG3* and *CTLA4*) (**Fig. S18E**). While CD8 T cells can also differentiate into a phenotypically distinct regulatory lineage and produce IL10 like Tr1 CD4 T cells in other diseases (118), DEGs during malaria in CD8 T cells subsets had limited overlap with DEGs that dominated the Tr1/Th1 response in CD4 T cells, and no evidence of increased IL10 expression was detected (**Fig. S18F-G**).”

Specific points:

For cytokine analysis the secretion of cytokines was blocked and cells were then stained for cytokines but are they stimulated ex vivo antigen-specifically? Or does they still produce cytokines ex vivo since they saw antigen recently in acute malaria?

Response: The reviewer's interpretation is correct - for cytokine analysis, PBMCs were cultured *ex vivo* with monensin to capture cytokines, but without stimulation. We interpret

this cytokine secretion being driven by antigen exposure during the blood stage of infection.

In line 227 the authors refer to a Figure 2G which is missing.

Response: This error has been corrected.

In line 670 the authors claimed that B cells are a major source of IL10. How they compared the amount of IL-10 produced by B cells to the amount of IL10 produced by other cell subsets?

Response: To compare the relative contribution of IL10 from B cells within all IL10 from the lymphocyte population, lymphocytes were identified by size, CD14-/live cells and then IL10+ cells analyzed by CD19/CD3/CD56 expression. A gating strategy is now presented in Supplementary Figure S22A and referred to in the results section. This data shows that IL10 from B cells is proportional similar to IL10 produced by T cells within the lymphocyte population.

I have difficulties to understand line 691 to 694. Why it is not possible to analyze data on an individual cell levels?

Response: This section has been edited for clarity and now reads: “However, once PBMC cells are subclustered, relatively low cell numbers were available for analysis to identify malaria DEGs in rare cell types. As such, we were not able to identify DEGs at the individual donor level, and therefore may have overlooked individual level heterogeneity” (lines 819-822). Cell number information for each individual/cluster is now included in Supplementary Tables S4, S7, S10, S13, S16 and S19 . Additionally, expression of all DEGs that were identified with all donor data at day0 and 28, are now presented at the individual level for all cell subclusters in Supplementary Figures S3, S5, S12, S14, S19 and S21.

REVIEWERS' COMMENTS

Reviewer #1 (Remarks to the Author):

The authors have addressed all concerns that I raised and I only have one minor comment.

line 701 "P-value calculated using Kruskal wallis and post-Dunn test (FDR adjusted for multiple, comparisons across the three cell types) indicated.

-Usually Dunn's posttest would already correct for multiple testing, no?

I would like to congratulate the authors to a very nice manuscript.

Reviewer #3 (Remarks to the Author):

The authors have substantially revised their manuscript, most significantly by boosting sample size in several experimental readouts. My concerns have been addressed both through the inclusion of new pieces of data and through modifying the text to note that possibilities other than those espoused by the authors could underlie aspects of the results presented. A few lingering concerns: slight typo line 396. Sentence on lines 524-527 is too strong as it is by no means clear that type I IFN is responsible for Tr1 programming in this study (as the authors state in lines 813-815). While the authors hint at a major limitation of their study being a lack of antigen (parasite)-specific B/T cell analysis, this should be made more clear on page 42/43.

Reviewer #4 (Remarks to the Author):

The author invest tremendous work to perform additional experiments and in rewriting the manuscript. The revised version clearly answered all my questions and concerns. I suggest to accept this manuscript for publication.

We thank the reviews for positive feedback. We have made final changes as outlined below.

Reviewer #1 (Remarks to the Author):

The authors have addressed all concerns that I raised and I only have one minor comment.

line 701 "P-value calculated using Kruskal wallis and post-Dunn test (FDR adjusted for multiple, comparisons across the three cell types) indicated.

-Usually Dunn's posttest would already correct for multiple testing, no?

I would like to congratulate the authors to a very nice manuscript.

Response: Post-Dunn test can be adjusted for multiple comparisons using different. Specific mention of p-value adjustments is required for Nature Communications editorial requirements, and therefore this information has been left in figure legend.

Reviewer #3 (Remarks to the Author):

The authors have substantially revised their manuscript, most significantly by boosting sample size in several experimental readouts. My concerns have been addressed both through the inclusion of new pieces of data and through modifying the text to note that possibilities other than those espoused by the authors could underlie aspects of the results presented. A few lingering concerns: slight typo line 396. Sentence on lines 524-527 is too strong as it is by no means clear that type I IFN is responsible for Tr1 programming in this study (as the authors state in lines 813-815). While the authors hint at a major limitation of their study being a lack of antigen (parasite)-specific B/T cell analysis, this should be made more clear on page 42/43.

Response:

Line 396 has been edited.

Line 524-527 has been edited.

We have added a specific line to discussion (line 839) to address limitation of antigen specific B/T cells.

Reviewer #4 (Remarks to the Author):

The author invest tremendous work to perform additional experiments and in rewriting the manuscript. The revised version clearly answered all my questions and concerns. I suggest to accept this manuscript for publication.